# Far-travelled 3700 km lateral magma propagation just below the surface of Venus

H. El Bilali [1] ✉ & R. E. Ernst [1] ✉

The Great Dyke of Atla Regio (GDAR) is traced for ~3700 km on Venus, as a surface graben (narrow trough) interpreted to overlie a continuous laterally-emplaced underlying mafic dyke (vertical magma-filled crack). The GDAR belongs to a giant radiating dyke swarm associated with Ozza Mons (volcano), Atla Regio plume, and was fed from a magma reservoir ~600 km south of the Ozza Mons centre. A 50-degree counter-clockwise swing of the GDAR at 1200 km from the centre is consistent with a 1200 km radius for the underlying Ozza Mons plume head, and a stress link to the 10,000 km long Parga Chasmata rift system. Our discovery of the GDAR, should spur the search for additional long continuous single dykes on Venus (and Earth), with implications for estimating plume head size, locating buffered magma reservoirs, mapping regional stress variation at a geological instant, and revealing relative ages (through cross-cutting relationships) over regional-scale distances.

Dyke swarms are the dominant mode of transport of mafic magma on planetary bodies including Earth, and the largest swarms have giant radiating or circumferential patterns that can be linked to mantle plume centres[1–9]. Radiating swarms are interpreted to be laterally emplaced from their plume centre regions whereas giant circumferential swarms are likely fed both vertically and in part laterally. The longest radiating swarms on Earth have a maximum known radius of 2000 km (for the 1270 Ma Mackenzie of northern Canada[2,3]) and perhaps up to nearly 3000 km (for the 201 Ma Central Atlantic Magmatic Province (CAMP) LIP[10]). It is interpreted in such cases that the magma is generated and ascends vertically in the plume centre region (potentially up to a distance of a few hundred km out from the plume centre), and then is laterally emplaced for the entire remaining distance[3,11,12].

Given this lateral emplacement interpretation, it can be predicted that individual dykes of these giant swarms must extend the length of the swarm (e.g. up to ~3000 km). However, the longest distance that individual dykes have actually been traced so far on Earth is much shorter: the 1140 Ma Great Abitibi dyke of the Canadian shield is traced for 700 km[13], a Red Sea dyke for about 1000 km[14], and the Great Dyke of Zimbabwe for 550 km[15]. On Earth it is difficult to track the full extent of individual dykes for at least three reasons: (1) a dense dyke swarm prevents continuous tracking of individual dykes, (2) continental breakup can fragment swarms, and (3) younger sedimentary or volcanic units can mask the distribution of dykes.

Due to the current hot surface temperature (ca. 450 °C), there is an absence of fluvial erosion (and wind erosion is relatively minor) on Venus e.g.[16–18]. extending back at least hundreds of myr (based on cratering age constraints of ~300–1250 Myr[18], cf.[19], or ~240 Ma[20], and back to about 700 Ma based on climate modelling[21]. As a consequence, the primary surface of geological units and structures is preserved except where covered by younger lava flows, obscured by superimposed tectonics, or covered by rare sand dunes or talus associated with rift zones.

The relatively minor role of erosion[17] means that intrusive units such as dykes are not exposed; however, their presence can be inferred from their surface expression as narrow grabens, and also pit chains and chains of shield volcanoes inferred to overlie blind dykes (dykes not reaching the surface) that are, for the most part, interpreted to be laterally emplaced[9,22–24]. There have been detailed studies of graben systems (interpreted in the context of dykes) in multiple regions of Venus[9,25–27] and other planetary bodies, notably Mars[4,6,28–30].

In the global reconnaissance study by ref. 22, 163 radiating systems were identified on Venus, with maximum radii ranging from 40 to ≥2000 km, and with an average radius of ~325 km. This study[22] used the Magellan SAR (synthetic aperture radar) data, which had been compressed once to produce images (with a resolution of 225 m/pixel[31]). Subsequent regional studies using the full resolution Magellan SAR images (~125 m/pixel) reveal a greater number of radiating systems.

[1]Department of Earth Sciences, Carleton University, Ottawa, ON, Canada. ✉e-mail: hafidaelbilali@cunet.carleton.ca; Richard.Ernst@Carleton.ca

For instance, 5 times as many were recognized in the Guinevere Planitia region where a 3800 km long radiating graben system was reported by[24], consisting of a 1000 km radiating portion focussed on Theia Mons, Beta Regio, that transitions outward into a linear 2000 km long N-S swarm[24]. Similarly, a radiating graben system with a radius of 6000 km has been associated with the 1200 km radius Artemis Corona magmatic centre[32]. In these cases, it is typically interpreted that these grabens overlie dykes which were emplaced laterally for up to these full distances, away from the magmatic centres (as on Earth, as discussed above). However, no single dyke of these swarms has been continuously traced for such distances. Herein, through detailed analysis using full resolution Magellan SAR images we track a single graben (interpreted as a dyke) for at least 3700 km.

Part of the BAT (Beta-Atla-Themis) region, Atla Regio is recognized as a major plume centre, on the basis of gravity, topography, geoid and other criteria favouring deep dynamic support, and being the locus of triple junction rifting with the largest rifts extending to the ENE, SW and SE to connect with other Regios, and also associated an radiating dyke swarm (Fig. 1 inset)[9,33–36].

In this work, we identify the longest individual graben (interpreted as overlying a dyke), traced so far, on Venus (and indeed in the solar system) and discuss the implication of this discovery, and more generally, the importance of identifying other long individual dykes.

## Results and discussion
### Tracing of the Great Dyke of Atla Regio (GDAR)
We have traced a single graben nearly continuously for 3700 km using the full resolution (~75–150 pixel[37,38]. left-look and right-look Magellan SAR images (Fig. 1, images 1 and 3). A 1700 km portion of this graben structure was previously mapped as Penthesilea Fossa (i.e. a long, narrow depression (trough), interpreted as a graben) in Taussig Quadrangle V-39[39]. Our mapping has confirmed the extent of this single graben, and also traced this graben west of Taussig Quadrangle back to the interpreted source associated with Ozza Mons. As with

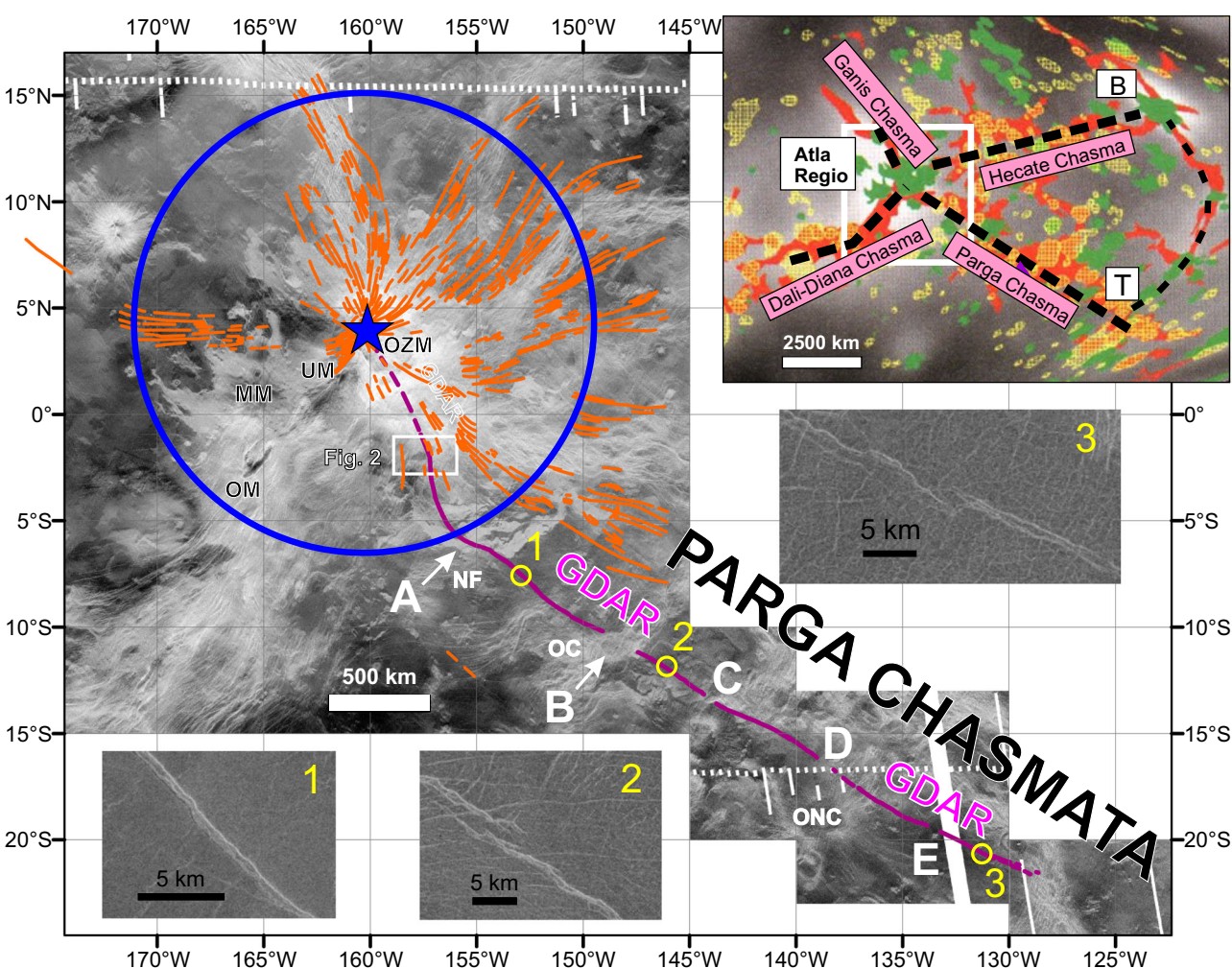

**Fig. 1 | Trace of Great Dyke of Atla Regio (GDAR).** The GDAR (solid purple line) can be traced for more than 3700 km from a source magma reservoir (and approaching 4300 km if ultimately fed from a source at the centre of Ozza Mons; dashed purple line). The GDAR is part of a radiating system of grabens (orange lines) interpreted to represent a giant radiating dyke swarm focussed on Ozza Mons (orange lines from[9]) with centre marked by blue star. The absence of orange grabens in the SW quadrant is due to younger superimposed volcanism and grabens of Maat Mons (MM) and Unnamed Mons (UM). Blue circle indicates the transition distance (1200 km) from the purely radiating pattern to a more distal trend that is influenced by regional stresses. At a distance beyond 1200 km the GDAR becomes parallel to Parga Chasmata (rift system) (see inset). OZM Ozza Mons, MM Maat Mons, UM Unnamed Mons, OM Ongwuti Mons, NF Ningyo Fluctus, OC Oduduwa Corona, ONC Onenhste Corona. White box locates Fig. 2. Letters A-E indicate important features discussed in the text. Examples of the appearance of the graben overlying the GDAR at different positions are shown with numbers linking them to positions along the GDAR. Location 1 (7.4 S, 206.9E), Location 2 (11.8 S, 213.8E), Location 3 (20.8 S, 229.1E). Inset diagram (after[9]) shows location of BAT region with major plume generated magmatic centres: Atla Regio, Beta Regio (B) and Themis Regio (T) connected by major rift systems (Chasmata), with background image after[63] showing major volcanic edifices (green), coronae (yellow) and rift zones (red) superimposed on the geoid (grey scale, lighter is higher).

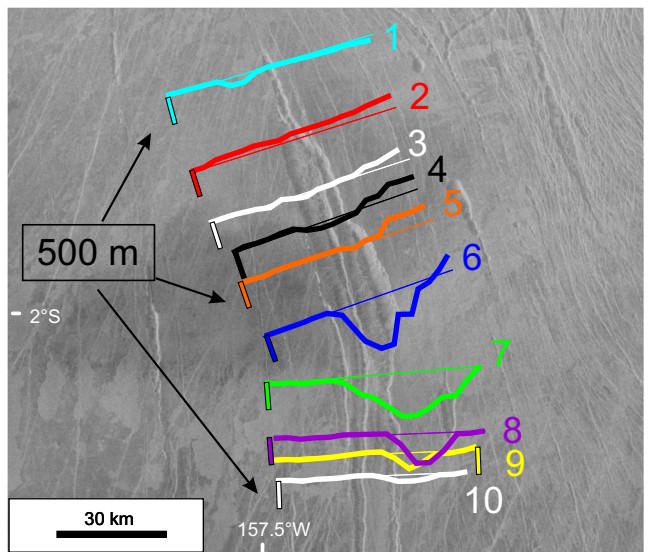
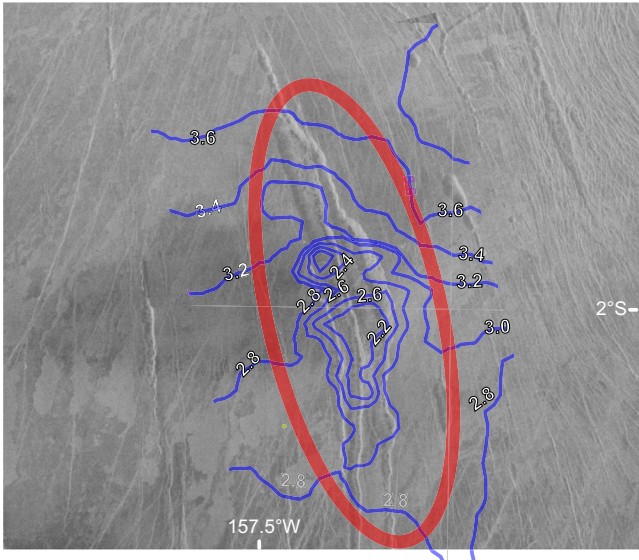

**Fig. 2 | Source of GDAR.** There is evidence for a trough at the beginning of the GDAR, which is proposed to represent roof collapse above a magma reservoir source for the GDAR. Topographic profiles (1-10) based on Magellan altimetry data (left-side image) and contours in 1000 s of metres (right-side image) indicate a trough extending from between profiles 5 and 6 and continuing through profile 9 with a maximum width of 18 km, length 72 km and maximum depth of about 900 m. The apparent eastward offset of the trough in profiles 8–10 may result from the known issue of up to -20 km uncertainty in the positioning of the Magellan altimetry data (e.g.[46]. See white box with location in Fig. 1. In each profile the vertical reference line is 500 m.

other grabens (as discussed above) we interpret this particular graben to represent a single mafic dyke.

However, given the significance of our proposed recognition of a single mafic dyke of such length, we also consider other origins for a long and narrow geological feature. Could this represent a fault (strike slip or thrust), or purely tectonic extensional structure? If it were a strike- slip fault then the boundaries of older host units should be offset across the feature, which is not observed. If it were a thrust fault then given the minor role of erosion (see discussion above) there should be a topographic change across the feature, which is also not observed. A purely extensional feature without a dyke underneath would imply regional extension, but the result would be rift zone, with a complex pattern of rifting[40], not a single narrow graben several thousand km long.

This leaves a dyke interpretation as the most likely. Another point to discuss is whether the dyke is fed vertically or horizontally. If fed vertically then this would require an underlying source co-extensive with the entire length of the dyke (and indeed co-extensive with the full extent of the Ozza Mons radiating swarm). On Earth, using flow fabric (magnetic fabric) and geochemical studies it has been shown that such large dyke swarms are mainly fed horizontally except near their plume centre source, where emplacement is vertical[3,11,41]. It is also typically interpreted that major radiating graben systems on Venus and Mars are also underlain by laterally emplaced dykes[5,6,22,23].

Our final point is that the underlying dykes must be mafic (mainly basaltic) in composition rather than intermediate or felsic. Two reasons are relevant, (1) nearly all the volcanism that extends across 80% of Venus is interpreted to be basaltic based on flow morphology (e.g.[42], and also from measurements by Soviet lander data[43]; so if dyke feeders are present they should dominantly be mafic rather than felsic; (2) more specifically there are many examples of flows with basaltic morphology, emanating from individual graben (e.g.[24]). Thus, we interpret that this 3700 km long graben (fossa) is a single mafic dyke (likely equivalent to a large dolerite-gabbro dyke on Earth) which we have termed the Great Dyke of Atla Regio (GDAR).

We note that there are some breaks and gaps in the continuity of the GDAR. Small breaks along this graben are also associated with en echelon offsets (e.g. detailed image 2 in Fig. 1) which are common in regional mafic dyke swarms on Earth and typically interpreted to represent a continuous laterally propagating dyke at depth, for which the upper portions ascend into slightly different stress regime causing local rotation and segmentation of the dyke[44].

Also, four gaps in the tracing of the GDAR were observed due to superimposed younger events: (1) a 213 km gap is covered by lava flows of Oduduwa Corona (from 10.160 S, 210.715E to 11.163 S, 212.501 E; location B in Fig. 1), (2) an intensely fractured radar bright zone (older or younger is not clear) obscures the GDAR for a distance of 166 km (from 13.240 S, 216.036E to 14.155 S, 217.430 E; location C in Fig. 1), (3) a 139 km gap is obscured by grabens (dykes) of Onenhste Corona (from 16.127 S, 220.963E to 17.090 S, 222.188E, location D in Fig. 1), and (4) a 66 km gap is obscured by radar dark material, likely dust from Felicia impact (or radar dark lava flows) (from 19.334 S, 226.177E to 19.627 S, 226.762 E, location E in Fig. 1). In each case, the continuity of the GDAR across the gap is clear (based on matching of trend and width of the graben). The last position to which the GDAR can be reliably traced is 20.878 S, 229.299E.

The width of the graben along the GDAR (measured using the Magellan SAR images) is typically $2 \pm 1$ km which using the scaling ratios of $3.5 \pm 0.5$ [after 6, using data from[45]], and with error propagation would translate to a depth to the top of the underlying dyke of $600 \pm 300$ m.

Topographic profiles using Magellan altimetry data across the graben reveal no discernable trough indicating that the maximum depth of the graben must be less than the vertical resolution (50-100 m) of the Magellan altimetry data[31,46]. The cross-section area of the surface graben should equal the rectangular area given by the distance from the top of the dyke to the surface, multiplied by the dyke width. Therefore, multiplying graben width ($2 \pm 1$ km) times inferred graben depth ($<75 \pm 25$ m) should equal the estimated depth to the top of the dyke ($0.6 \pm 0.3$ km) times the dyke width. Using the above numbers, and with error propagation, the dyke width can be determined as $<250 \pm 195$ m. Obviously, there is considerable uncertainty in our estimate of dyke thickness, but we note that the range of values overlaps with measured widths (20 to >200 km) for terrestrial mafic dykes associated with mantle plumes[2,10]. Also, our outlined approach will yield better estimates (of the depth to the top of the dyke and dyke

width) using the expected higher resolution altimetry of future Venus missions.

## Link to Ozza Mons

The GDAR is part of the giant radiating swarm focused on Ozza Mons (Fig. 1 ref. 9), and is therefore linked to Ozza Mons. The GDAR appears to begin at a NNW-trending steep-sided elongate trough basin (72 km long, 18 km wide and up to 900 m deep) starting ~600 km from the Ozza Mons centre (Fig. 2) (see discussion below).

## Control on trend of dykes

An individual dyke represents a single lateral emplacement event potentially emplaced in hours, days or weeks[1,6,45,47,48]. Recognition of such a long continuous dyke provides a rare opportunity to understand regional stress variation across a large region of Venus, at a geological instant. Regional mafic dyke swarms are emplaced as Mode 1 fractures, with trend perpendicular to the minimum compressive stress, and parallel to the maximum compressive stress[47–50]. They can have giant radiating and circumferential patterns associated with mantle plumes[8]. For radiating swarms, the radius of the domal uplift is linked to the size of the radiating portion of the swarm. However, given continued magma supply, the radial swarm will continue to propagate outward but then swing into a regional stress trend beyond the area of domal uplift[3,51].

## Swing in trend at Ningyo Fluctus

The GDAR trends initially to the SSE, then, at a distance of about 1200 km from the plume centre (blue star in Fig. 1) swings 50 degrees counter-clockwise (over a distance of about 300 km) when crossing Ningyo Fluctus (location A in Fig. 1). This change in trend is consistent with similar swings in other parts of the Ozza Mons radiating swarm[9], and is interpreted to indicate a plume head radius for Ozza Mons of 1200 km[9] (blue line in Fig. 1). The criteria for interpreting the transition from fully radiating to a swing in trend, as marking the edge of the underlying mantle plume, is discussed in[9,52]; it is linked to the extent of domal uplift associated with flattening of the plume head against the lithosphere. This approach derives from terrestrial literature on using a radiating dyke swarm to interpret plume head size, first done by[3] for the interpretation of the outer edge of the 1270 Ma Mackenzie plume head in northern Canada.

## Consistent SW trend parallel to Parga Chasmata

Beyond 1200 km, the GDAR has an approximately linear trend which is parallel to the trend of the 10,000 km long rift system of Parga Chasmata. This indicates that the plume-generated uplift of Ozza Mons (which produced the radiating swarm including the GDAR) was coeval with active extension along the 10,000 km Parga Chasmata—at least for the 3000 km that is paralleled by the GDAR. This is reasonable since Parga Chasmata represents one arm of the 4-armed 'triple' junction rifting associated with Ozza Mons (Fig. 1 inset). Previous studies of local radiating swarms along Parga Chasmata had recognized some with trends locally swinging into parallelism to the Parga Chasmata rift zone[22,27,53], but the remarkable continuity of the GDAR confirms a link between the timing of Ozza Mons (and its >3700 km long GDAR) and active extensional stresses along Parga Chasmata as a whole.

## Geologically instantaneous regional time marker

The continuity of the GADR allows its use as a precise time marker to indicate relative ages along its length, based on cross-cutting relationships. The GADR appears to cut all geological units, both those associated with Ozza Mons, and along its path parallel to Parga Chasmata, except in three cases mentioned above: the GDAR is obscured by lava flow units on the north side of Oduduwa Corona (as mentioned above) indicating that Oduduwa Corona is younger (location B in Fig. 1). Multiple grabens of the radiating dyke swarm of Onenhtse

Corona crosscut the GDAR, indicating that Onenhtse Corona is also younger than the GDAR (location D in Fig. 1). Radar dark (and hence fine-grained) material from Felicia impact crater obscures the dyke at location E in Fig. 1.

## Characterizing the magma reservoir source

Given the current lack of significant erosion on Venus at the time of the GDAR (as discussed above), the surface expression of dykes as graben-fissure lineaments is visible unless covered by younger flows. A closer look at the entire length of the GDAR using the high-resolution Magellan SAR images (apart from the gaps mentioned above), confirms that nowhere along its 3700 km length did any significant lava spill out of the GDAR. We cannot rule out very small insignificant flows emanating from the dyke, having areal extents on the order of 6000–22,000 m$^2$ on the order of a single Magellan SAR pixel (75–150 m[37,38]). Therefore, our conclusion is that the entire volume (or at least essentially the entire volume) of magma injected laterally into the GDAR is still preserved within the dyke.

We estimate the volume of magma injected into the GDAR, by multiplying dyke length by its width and vertical extent. A key question is the vertical extent of magma in a dyke. Models for dyke injection indicate that the dyke is 'floating' in the crust at a level of neutral buoyancy that matches the cumulative density of the vertical blade of mafic dyke magma against that of the host rocks over the same vertical distance[6,10,47,54]. An earlier estimate by[2] suggested that such laterally-emplaced terrestrial dykes bottom out in the lower crust. As summarized in[47] the maximum vertical height of a terrestrial dyke is likely between 10 and 20 km, although the analysis of[6] suggests vertical extents of >50 km. Estimates for the thickness of Venusian crust vary, and average values typically vary between 8 and 25 km[55–57], with larger values for highlands e.g.[56]. In the calculation below we use both 50 and 20 km for the GDAR's vertical extent. Widths of dykes in major regional swarms on Earth can range between 20 and 60 m, but the largest can reach more than 200 m[6,10]. Based on these values, the maximum and minimum volumes of the GDAR would be: 37,000 km$^3$ (=3700 km × 200 m × 50 km) and 1480 km$^3$ (=3700 km × 20 m × 20 km).

These values are compared with a very approximate estimate of the volume of the interpreted source magma reservoir, located about 600 km south of the Ozza Mons plume centre. We hypothesize that this steep sided trough basin (1200 km$^3$ = 72 km long × 18 km wide × 900 m deep) (Figs. 1 and 2) represents roof collapse above an underlying dyke-like magma reservoir. On Earth some radiating swarms have been linked to lateral emplacement from major intrusions[3], and on Venus we are also identifying source magma reservoirs through topographic depressions marking roof collapse as magma is expelled as flows or dykes (e.g in[58]). Lateral emplacement of the entire extent of the GDAR from the proposed reservoir would require multiple cycles (of filling and emptying of this reservoir). Using the above volume estimates for the GDAR this would suggest between 1 and 37 cycles (1480 km$^3$/1200 km$^3$ to 37,000/1200) to emplace the GDAR. This would indicate that the shallow magma reservoir (marked by the trough basin) was likely buffered with regular input of magma from another reservoir[59], at greater depth or perhaps closer to the centre of Ozza Mons (Fig. 1).

## Lessons for long dykes in other radiating swarms on Venus

The implications of our discovery of the 3700 km long Great Dyke of Atla Regio, should spur the search for additional long continuous single dykes on Venus, with implications for plume head size, buffered magma reservoirs, mapping regional stress variation across a large region of Venus at a geological instant, and revealing relative ages (through crosscutting relationships) over regional distances. Recognition of the GDAR and other long dykes will be valuable as observation targets for the new Venus missions planned over the next couple

of decades including (VERITAS, DAVINCI, EnVision, Venera-D, Shuk-rayaan-1, VOICE).

On Venus, such long dyke swarms are particularly likely in the other major Regios on Venus that are interpreted to mark major plumes[33–35,60], such as Beta, Bell, Eistla, and Imdr regios.

## Lessons for dyke swarms on Earth

The radius of the plume head of Ozza Mons (1200 km) is in line with modelling, tomography, and dyke swarm radii for plume head size on Earth[61], and supports the matching of plume head characteristics between Earth and Venus. Therefore, the great length of the GDAR indicates that terrestrial LIPs could have individual dykes of similar length. Thus the long 2800 km swarm of the East North America (ENA) portion of the CAMP LIP could include individual dykes of that length. This is an important conclusion since the difficulty of tracing individual dykes leads to much shorter (and incorrect) estimates of maximum individual dyke lengths for LIPs, typically on the order of 10s of kms. However, lateral emplacement indicates that the individual dykes of a radiating swarm should track back to a magma reservoir source near or at the plume/diapir centre, with implication for much longer dyke lengths.

On Earth giant radiating swarms are observed in continental crust, but have so far not been recognized in Earth oceanic crust. However, since the composition of Venusian crust appears mainly basaltic (in contrast to the major continental area on Earth that host giant dyke swarms) then this result from Venus suggests that the tracing of such long individual dyke swarms could also be associated with terrestrial oceanic plumes/oceanic plateaus cutting basaltic oceanic crust.

## Methods

We used full-resolution (-125 m/pixel) Magellan SAR images and its altimetry data from USGS Astrogeology Science Center (https://astrogeology.usgs.gov/search?pmi-target=venus) and mapped in Arc-GIS ArcMap v. 10.3. JMARS (Java Mission-planning and Analysis for Remote Sensing)[62] was used for reconnaissance and generation of topographic profiles.

## Data availability

The Shape Files shown in this manuscript are available as a Supplementary File, and include the generalized linework for the Ozza Mons radiating graben system (Fig. 1) and the detailed tracing of the Great Dyke of Atla Regio (GDAR) (Fig. 1).

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

## Acknowledgements

We thank Ken Buchan, James Head and Ross Mitchell for valued discussions. This research is funded by NSERC Discovery Grant RGPIN-2020-06408 to R.E.

## Author contributions

H.E. and R.E. both developed the concepts. H.E. performed the detailed mapping. Both H.E. and R.E. wrote the initial draft and revised and finalized the manuscript.

## Competing interests
The authors declare no competing interests.
