## [Peer Review File · Nature Communications]

Far-Travelled 3700 km Lateral Magma Propagation Just
Below the Surface of VenusReviewers' comments:

Reviewer #1 (Remarks to the Author):

Review of "The Great Dyke of Atla Regio, Venus: The Longest Traced Individual Dyke in the Solar System" for Nature Communications. April 29, 2023

In this study, the authors traced surface graben on Venus for over ~3700 km. They claimed that these graben are the surface manifestation of a single dyke that is continuous, except for several breaks (the largest of which is ~200 km). They inferred that this "continuous laterally-emplaced underlying mafic dyke (vertical sheet of magma)" was "fed from a buffered magma reservoir" located south of a putative mantle plume at Ozza Mons. They discussed implications of this conclusion for understanding the regional volcanic and tectonic activity. My comments below are quite critical, so I want to emphasize first that this study is very interesting. Venus is a superlative setting to study volcanic and tectonic processes. The conclusions of this manuscript, if convincingly established, would have many important implications. However, the stronger the conclusions, the stronger the scrutiny that a reviewer has a duty to apply...

Unfortunately, I recommend rejecting this manuscript because virtually all the major conclusions are not convincingly established—and, in fact, many of them seem to be in doubt. Overall, the manuscript relies on qualitative analogies (and arguably excessive self-citations) to establish its conclusions, which are not supported here by other types of data analysis or substantial quantitative modeling. Many statements in the manuscript merit heavy caveats. Finally, the proposed acronym ("Great Dyke of Atla Regio (GDAR)") and epithet ("longest traced individual dyke in the Solar System") seem over-hyped. As discussed below, the GDAR has **not** been established as continuous—and the manuscript acknowledges multiple times that there could be longer dykes on Venus and elsewhere.

I elaborate on my concerns in order of appearance in the manuscript:

- Line 13: The manuscript claims that the dyke is continuous, laterally-emplaced, and mafic. However, the manuscript does not include sufficient evidence to prove these claims. The graben that allegedly are the surface manifestation of the GDAR are **not** continuous. There are multiple breaks, one as large as ~200 km. The manuscript claims (lines 141–152) that, because the dyke is allegedly continuous, it must be older than Oduduwa Corona. However, this seems like circular logic? If the GDAR is not continuous (e.g., if it were multiple dykes of different ages that happen to line up), then we cannot use cross-cutting analyses as simple as those presented here. Finally, the claims that the dyke was laterally-emplaced and (especially) mafic seem to rely on qualitative analogies to dyke swarms on Earth. But there are, in principle, ways to identify the composition of rocks besides drawing lines on a map. I do not believe that the analyses in this manuscript uniquely constrain the putative magma to a mafic composition.

- Lines 18–19: The last sentence of this abstract relies on a very simple calculation that has substantial uncertainties that are not addressed (discussed toward the end of my comments). I think it is hazardous to the unwary reader to have things stated as fact when they are quite speculative.

- Line 57: The “at least back to about 1 Ga” claim is wildly overstated. The surface might not be nearly that old, on average. The canonical Venus II book chapter by McKinnon et al. (1997) estimate the surface age as ~300–1250 Myr. Le Feuvre & Wieczorek (2011, Icarus) derived a cratering age of ~240 Myr for the surface of Venus. So, giving a number like 1 Ga with total certainty might mislead readers. Furthermore, the citation here is inappropriate—and is, alas, an example of this manuscript’s habit of self-citation. Reference 17 by Way, Ernst, & Scargle is a very interesting study, but it focuses on the timing of large-scale volcanism on Earth. It does not report, as far as I can tell, substantial new analyses of the age of the surface of Venus or possible rates of erosion. I recommend citing works that establish the stated claims (in this case, after modifying the claim appropriately). Finally, “significant” is doing a lot of work in this sentence. There is some sedimentary activity on Venus today. Venera landers observed stuff moving; there are dunes; craters produce sediment that is clearly moved around over time; etc. What counts as “significant”?

- Lines 66–67: The resolution of the SAR images is overstated throughout the manuscript. The available mosaics are presented at a pixel scale of 75 meters per pixel, but they are over-sampled. The real resolution is ~125 meters per pixel (e.g., Saunders et al. 1992).

- Lines 72–74: The hypothesis that the GDAR is laterally-emplaced is one of the major claims of this manuscript. However, again the main evidence seems to be qualitative analogy to other locations on Venus and Earth. Specifically, how “can [it] be inferred” that individual dykes are laterally emplaced? How was this done for the radiating swarm around Artemis? Can those same methods be applied here? Incidentally, this is also the place in the manuscript that highlights why calling the GDAR “longest” might seem like puffery—dykes around Artemis could be much longer.

- Lines 80–81: The claim that “Atla Regio is recognized as a major plume centre” needs a reference (not a self-citation, please) and elaboration. On what basis do folks infer the presence of a mantle plume? Are there competing hypotheses?

- Line 98: “72 km”?

- Line 106–107: “The recognition of the GDAR” is stated as a fact here (and elsewhere) and used to support many further claims. However, the “GDAR” has several breaks in continuity, and the authors have not mustered sufficient proof that these breaks are not real gaps between different dykes. In any case, all these statements need caveats. If the GDAR is one continuous dyke, then we can make certain claims.

- Line 126: “Confirming” seems way too strong here. Maybe “consistent with...”? Determining the plume head size would probably require geophysical data like seismology, high-resolution gravity, etc.

- Line 137: Again, “the remarkable continuity of the GDAR provides a conclusive link” is way too strong. It’s not entirely continuous.

- Lines 142–152: Again, the topic sentence of this paragraph is way too strong. Also, “GADR”? The dyke

could be used as a precise time marker if it were continuous. However, that seems to be an *assumption* because there is no specific proof that the breaks in visible continuity do not correspond to the real end of a dyke. Line 145 says that “the only significant break in the continuity of the GDAR is for 200 km” but “significant” is in the eye of the beholder (and I believe disallowed in Nature journals unless one has performed a statistical test with a quantitative threshold). Continuity is the sine qua non of this manuscript, so I believe that proving the stated conclusions would require evaluating every single break in continuity observed along the trace.

- Lines 160–161: “the surface expression... is visible unless covered by younger flows” implies that the depth of the dyke does not change along its length, correct? If the dyke were vertically emplaced or laterally emplaced but at different depths along track, then the surface expression could change even if the surface were everywhere the same age?

- Lines 162–165: Apologies that I sound like a broken record, but I do not believe that it is possible to draw such a strong conclusion from the evidence presented in this manuscript. Specifically, the authors state “nowhere along its 3700 km length did any lava flow spill out of the GDAR. Our conclusion is that the entire volume of magma injected laterally to form the GDAR is still preserved within the dyke.” Let me draw a qualitative analogy. Baltis Vallis is the longest lava (almost certainly lava) channel on Venus. However, scientists debate whether lava flowed from north to south or from south to north. That’s because it’s not obvious where lava spilled out of Baltis Vallis. However, I would not thus conclude that all the lava that formed Baltis Vallis is *still inside* Baltis Vallis. Our available data are limited (e.g., the radar images have low-resolution; we do not have compositional data; we haven’t yet done sub-surface radar sounding that could identify individual flows). I could make similar arguments about over a hundred other lava channels on Venus. So, can the authors prove that *their failure to observe* lava spilling out of the dyke means that *no lava at all* spilled out of the dyke? Surely there is some amount of lava (maybe a huge amount) that could have spilled out but escaped detection? Maybe EnVision and/or VERITAS will find some? In an extreme case, perhaps nearly the *entire* volume of lava spilled out of the end of the dyke, but then the end of the dyke was covered by a younger flow? Overall, the stated conclusion seems like an extreme upper bound on reality—and the manuscript should acknowledge other possibilities.

- Lines 165–184: Here is the quantitative part of the manuscript, which underlies the central conclusion that the GDAR was fed from a buffered magma reservoir. However, once again, this conclusion depends on an assumption that was not well-justified. The calculation of the magma volume includes an uncertainty analysis related to measurements of the width of the dyke. However, the vertical extent of the dyke is even more uncertain—but no such uncertainty is included in the calculation. Specifically, the manuscript uses 20 km as the vertical extent. If the vertical extent is <10 km, then only one cycle of magma from the reservoir is (maybe) required to fill the dyke—meaning that the conclusion of a “buffered magma reservoir” goes away. How do we know that the vertical extent is not <10 km? Cited references 33 and 9 seem to focus on Mars and Earth. Reference 33 admits vertical heights <10 km—and conditions on Venus could be different than on Earth. In particular, the relevant calculation would require assumptions about the density of the magma and the host rocks. The manuscript does not provide estimates for those densities, which must be considered uncertain in the absence of much relevant geophysical data for Venus. Similarly, one number is used for the volume of the volume of the

source magma reservoir, but that number is uncertain. As with virtually all the conclusions in this manuscript, the scenario described by the authors seems *possible* but is by no means the only possibility that is consistent with the data.

- Lines 197–201: To support transparent and reproducible science, I believe that the authors should make available, in an accessible digital format, all the data used to generate the figures in the manuscript (e.g., all traces and topographic profiles).

- References. By my count, almost half of the citations are self-citations, which seems a bit excessive. I recommend reconsidering these self-citations. As with the “1 Ga” “[17]” example discussed above, there might be many cases where it would be more appropriate to cite a paper that directly performed the relevant analysis, rather than a study, which included the senior author, that simply mentioned the issue.

Reviewer #2 (Remarks to the Author):

Review of Nature Communications Manuscript: The Great Dyke of Atla Region, Venus: The Longest Traced Individual Dyke in the Solar System

General comments:

This manuscript presents a strong case for identifying the longest single dyke in the Solar System on the planet Venus. This identification could have extremely important implications for our interpretation of stress regimes on Venus, as well as our ability to determine relative ages across large regions. This manuscript is well written and lays out its case carefully. However, it would benefit from some slight restructuring and some additional clarification to better support its argument.

Specific Comment:

The authors frequently say that they are observing (or mapping, or measuring) dykes when what they really mean is that they are observing graben that they are interpreting to overlie dykes. This is a pretty important distinction to make. The papers the authors are citing to support their interpretation of dykes beneath the graben all assume that it is the emplacement of the dykes that forms the overlying graben. There is still quite a considerable amount of debate about this issue. For all of the literature that supports the contention that graben form above dykes, there is an equal number of papers showing that dykes don't necessarily form overlying graben. Not every graben on Earth has a dyke beneath it, so it is odd that planetary researchers seem to assume that there is a dyke beneath every graben on Venus (or Mars). And there are multiple locations on Earth where exposed cross-sections of dykes show no fracturing (or deformation of any kind) above them. Physical models have also shown that dyking does not necessarily form graben.

To be fair, the authors do state that they are making this interpretation in several places in the manuscript. But it doesn't make it any less jarring when they then talk about measuring the length of a

dyke when what they really mean is they measured the length of the graben that they are interpreting to overlie a dyke. And to be clear, I am not trying to suggest that there are no dykes beneath the graben radial to Ozza Mons. I just think that the authors could be more precise and accurate in their writing. Such clarity would only strengthen their arguments and support their case for the presence of a 3700 km long dyke on Venus.

Minor Comments:

I have made some suggestions in an attached annotated PDF that I believe would strengthen the paper.

Reviewer #3 (Remarks to the Author):

In this short paper, El Bilali & Ernst analyze Magellan radar images of Venus' Atla Regio to map a giant dyke that is described as the longest dyke traced so far on Venus and the entire solar system. The authors demonstrate a length of least 3700 km and potentially almost 4500 km, which substantially exceeds the longest dyke traces on Earth. The authors trace the dyke based on its graben signature and use elevation data to identify major troughs. I found this letter relevant and interesting to read and it is relatively easy to follow the manuscript even without being an expert on dyke swarms and/or geological mapping. In this sense, I feel this work is of interest for publication in a journal such as Nature Communications.

I have not spotted any major flaw in the way the results are obtained and interpreted. However, I was a little disappointed about the discussion of broader implications and significance of the detection of such a long dyke. Some issues that I would have liked to see discussed a bit more:

- 1.) Could such a long dyke imply that Venusian dykes may be longer than on Earth's or does it just demonstrate that terrestrial dykes are much more overprinted and we can trace their original length? Are there implications of such a long dyke on the properties and state of the interior (which are thought to be different and probably have been for substantial time for Earth and Venus)? For instance, could the place constraints on local crustal thickness and strength?
- 2.) What is a feasible tectonic regime that may support such long dykes? Is this more in line with classical end-members such as the stagnant or mobile lid tectonics, or could that be better explained by some more recent suggestions for Venus' tectonic regime, such as the plutonic-squishy lid?
- 3.) I understand that the 1200-km-distance at which the directional swing of the dyke occurs is here used as a proxy for the source plume head below Ozza Mons, but I would have liked a short discussion whether this indicates an unusually large plume or not, perhaps by comparison to better-studied terrestrial settings. In other words, is a very long dyke indicative of an unusually large or hot plume head? Or is the length of dyke swarm thought to be controlled mostly by other factors?

4.) The last paragraph (line 190++) points towards searching for other long continuous dykes on Venus. But what are regions on Venus which may display such features more likely than others? Also, in what sense may be upcoming Venus missions – Veritas, DaVinci, and EnVision – help to identify more such dykes and/or provide more details about the structure of the GDAR and its implications?

I think, some elaboration of points 1.) - 4.) would improve the manuscript and its impact on the Venus' science community the most, but I do have some additional minor comments, following below. Overall, I think this paper can be made suitable for publication with minor-to-moderate revision.

Additional comments:

Line 11++: The abstract lacks a statement of the broader significance for our understanding of either dykes or of Venus' tectonics. It would be good to add one if the word limit allows.

Line 17: Remove the comma after "1200 km".

Line 29: "It is inferred ..." Inferred based on what? Also, the word "inferred" seemed to appear quite often throughout the manuscript. Please check and think about alternatives.

Line 40/41: Where has been the longest dyke in the solar system before your detection of GDAR? Readers may wonder about this (well, I did ...).

Line 57: "...at least back to about 1 Ga ..." Why 1 Ga? Is that supposed to be linked to Venus' mean surface age? This is unknown and, yes, 1 Ga is in the range of estimates. But at the very upper end, and the most recent estimates tend to suggest younger surface ages perhaps as young as 100-200 Myr. Please clarify, why you write "at least 1 Ga" if actually have very little clue of how Venus' operated at this time (it may or may not had more active tectonics etc.)

Line 59: "... inferred to overlie blind dykes ..." I do not understand how such blind dykes can be recognized from radar images such as those used here? Would that not need some rather high-resolution gravity data at least? Or is this inference actually just used in analogy to terrestrial dykes for we may know about these blind dykes?

Line 64: I am puzzle by the word "once-compressed". Could it be also twice-compressed? Does it effectively mean that the compressed images have coarser resolution (225m/px) than the uncompressed (75m/px)?

Line 69: "In this study [20] ..." should be "In that study [20] ..."?!

Line 185++: This short paragraph falls a bit short compared to the others. Would you expect similarly long dykes extending outwards from Ozza Mons in various directions (at least until the 1200 km-limit up to which the regional stresses seem to be less significant)? Or could there be some preferential directions? In Figure 1, it seems that there is at least one quadrant (SW) of the blue circle without any radiating graben systems (orange line)? Why?

Line 203++: DOI's seem to be missing consistently in the reference. These are always useful to have. It seems references 4 and particularly 30 are earlier (reduced) versions of the contents presented here, so consider deleting these ref's, especially as they are conference abstracts without peer-review.

Figure 1: I am not sure what the color coding in the large-scale inset (upper right corner of the figure) is meant to represent!?

Figure 2: It was not clear to me immediately what the 500 m label in the left panel is supposed to mean. While I got it on a second look, consider adding a note in the caption to make it clear right away. Also, it seems that you show here a ~100-200 km long section from the beginning of the GDAR, that is close to Ozza Mons, I suppose. It would be interesting to see, for comparison, how this compares to a section further away, perhaps at around the 1200-km boundary or even further out. This could potentially indicate structural changes along the dyke

REPOSE TO REVIEWER COMMENTS (in red font)

Reviewer #1 (Remarks to the Author):

Review of “The Great Dyke of Atla Regio, Venus: The Longest Traced Individual Dyke in the Solar System” for Nature Communications. April 29, 2023

In this study, the authors traced surface graben on Venus for over ~3700 km. They claimed that these graben are the surface manifestation of a single dyke that is continuous, except for several breaks (the largest of which is ~200 km). They inferred that this “continuous laterally-emplaced underlying mafic dyke (vertical sheet of magma)” was “fed from a buffered magma reservoir” located south of a putative mantle plume at Oza Mons. They discussed implications of this conclusion for understanding the regional volcanic and tectonic activity. My comments below are quite critical, so I want to emphasize first that this study is very interesting. Venus is a superlative setting to study volcanic and tectonic processes. The conclusions of this manuscript, if convincingly established, would have many important implications. However, the stronger the conclusions, the stronger the scrutiny that a reviewer has a duty to apply...

Unfortunately, I recommend rejecting this manuscript because virtually all the major conclusions are not convincingly established—and, in fact, many of them seem to be in doubt. Overall, the manuscript relies on qualitative analogies (and arguably excessive self-citations) to establish its conclusions, which are not supported here by other types of data analysis or substantial quantitative modeling. Many statements in the manuscript merit heavy caveats.

Finally, the proposed acronym (“Great Dyke of Atla Regio (GDAR)”) and epithet (“longest traced individual dyke in the Solar System”) seem over-hyped.

REE: the use of the term “Great Dyke...” is common on earth for individual mafic (-ultramafic) dykes that are particularly long and significant. The text refers to examples such as the Great Dyke of Zimbabwe, and the Great Abitibi dyke. Other examples can be added if necessary.

REE: perhaps we have over hyped this--- but it is certainly the longest TRACED individual dyke in the Solar System—I am very knowledgeable on the terrestrial and planetary literature on dykes, and there is no longer individual dyke yet traced. So the hype is justified, but we have toned it down. In particular we have changed the title to:

“FAR-TRAVELLED (4000 KM) MAGMA PROPOGATION JUST BELOW THE SURFACE OF VENUS”

As discussed below, the GDAR has *not* been established as continuous—and the manuscript acknowledges multiple times that there could be longer dykes on Venus and elsewhere.

REE: We explain below why the interpretation of a continuous dyke is correct, and our explanation for gaps. We certainly indicate that the careful tracing of dykes in other swarms may yield longer dykes. But until that work yields a longer individual dyke, then the GDAR remains the record holder for the longest TRACED individual dyke.

The Artemis radiating swarm of (Hansen and Olive 2010 Geology) is the largest radiating swarm with the swarm extending up to 6000 km away from the Artemis centre. It is sensible to consider that these have been laterally emplaced for the entire 5000 km distance from the edge of corona ring (at about 1000 km radius) to the full extent of swarm—and thus 5000 km long individual dykes can be expected—but this will need to be confirmed by detailed mapping of the type we have done. Thus, our approach is also a template for how to approach identifying other very long individual dykes

I elaborate on my concerns in order of appearance in the manuscript:

- Line 13: The manuscript claims that the dyke is continuous, laterally-emplaced, and mafic. However, the manuscript does not include sufficient evidence to prove these claims.

REE: the continuity of the GDAR is based on its distinct width and appearance (see images in Figure 1) which allow it to be traced reliably across such gaps. We would be happy to 'walk' with the reviewer (using Magellan images) showing that our tracing of this dyke is unambiguous. We also noted in the text that the 1700 km long Penthesilea Fossa portion was previously recognized as a single dyke in Brian et al (2005).

The graben that allegedly are the surface manifestation of the GDAR are *not* continuous. There are multiple breaks, one as large as ~200 km. The manuscript claims (lines 141–152) that, because the dyke is allegedly continuous, it must be older than Oduduwa Corona. However, this seems like circular logic? If the GDAR is not continuous (e.g., if it were multiple dykes of different ages that happen to line up), then we cannot use cross-cutting analyses as simple as those presented here.

REE: Our logic is partly based on extensive terrestrial experience mapping dyke swarms including the 700 km long Great Abitibi Dyke for my PhD (Ernst and Bell 1992 J. Petrology). The GDAR is so distinctive in its graben width, that it is not reasonable to hypothesize multiple separate dykes (with similar wide grabens) just happening to line up.

Finally, the claims that the dyke was laterally-emplaced and (especially) mafic seem to rely on qualitative analogies to dyke swarms on Earth.

REE: The interpretation that regional dyke swarms on Earth (and Venus and Mars) are mainly laterally emplaced is so well established in the literature (as referenced in the manuscript); it is not really open to debate any more. We included some of the key papers, but could add at least 20 more in support lateral injection of regional dyke swarms on Earth and Venus and Mars.

But there are, in principle, ways to identify the composition of rocks besides drawing lines on a map. I do not believe that the analyses in this manuscript uniquely constrain the putative magma to a mafic composition.

REE: On Earth the major regional dyke swarms are dominantly mafic. Dykes of ultramafic (e.g. kimberlite dykes) and silicic composition tend to have much narrower widths. Given the similar regional scale of interpreted dyke swarms on Venus and Mars it is entirely reasonable to also infer their mafic compositions. Modelling by Wilson and Head (2002 JGR) also supports mafic compositions. In fact, there is no literature that I am aware of that interprets the dykes underlying graben (on Mars and Venus) to be anything other than mafic in composition.

REE: A further point: On Venus, there is a consensus that about 80% of the planet is covered by basaltic volcanism and the only potential area of silicic magma is in the older tesserae areas. In some cases, graben (interpreted to overlie dykes) are observed to feed flows (that are interpreted to be basaltic), and for these examples the dykes (feeding the flows) must also be mafic in composition.

- Lines 18–19: The last sentence of this abstract relies on a very simple calculation that has substantial uncertainties that are not addressed (discussed toward the end of my comments). I think it is hazardous to the unwary reader to have things stated as fact when they are quite speculative.

REE: Agreed. The calculation has significant uncertainties, which we discuss more completely in the text, but the observation of the remarkable topographic trough, and its interpretation as a collapsed magma chamber at the start of the GDAR--- is consistent with the abundant literature linking the source of regional radiating dyke swarms to such buffered magma chambers. We have revised the text to consider the range possible refilling values, and have add more justification of the values used for the calculation.

- Line 57: The “at least back to about 1 Ga” claim is wildly overstated. The surface might not be nearly that old, on average. The canonical Venus II book chapter by McKinnon et al. (1997) estimate the surface age as ~300–1250 Myr. Le Feuvre & Wieczorek (2011, Icarus) derived a cratering age of ~240 Myr for the surface of Venus. So, giving a number like 1 Ga with total certainty might mislead readers. Furthermore, the citation here is inappropriate—and is, alas, an example of this manuscript’s habit of self-citation. Reference 17 by Way, Ernst, & Scargle is a very interesting study, but it focuses on the timing of large-scale volcanism on Earth. It does not report, as far as I can tell, substantial new analyses of the age of the surface of Venus or possible rates of erosion. I recommend citing works that establish the stated claims (in this case, after modifying the claim appropriately). Finally, “significant” is doing a lot of work in this sentence. There is some sedimentary activity on Venus today. Venera landers observed stuff moving; there are dunes; craters produce sediment that is clearly moved around over time; etc. What counts as “significant”?

REE: We thank this reviewer for this useful and correct point. 1 Ga was added as a shorthand for the timing, but we have revised the text to include the suggested new references (McKinnon et al. (1997) and Le Feuvre & Wieczorek (2011, Icarus), to better reflect the uncertainties in timing.

To add these uncertainties to the timing of the formation of the Venusian surface. we also note that there is an emerging consensus that the Venusian surface formed over a range of time in a number of separate, not in a single volcanic resurfacing event.

REE: The word significant is used in four places in the manuscript and given Nature’s policy of restricting its use to statistical cases, then we have changed the word in each of the four cases.

- Lines 66–67: The resolution of the SAR images is overstated throughout the manuscript. The available mosaics are presented at a pixel scale of 75 meters per pixel, but they are over-sampled. The real resolution is ~125 meters per pixel (e.g., Saunders et al. 1992).

REE: minor point—and in fact the actual resolution varies across the planet (because of the changing distance to the surface in each pass of the Magellan spacecraft), but we indicate that the resolution varies across the planet but with an average of 125 m – as suggested by this reviewer.

- Lines 72–74: The hypothesis that the GDAR is laterally-emplaced is one of the major claims of this manuscript. However, again the main evidence seems to be qualitative analogy to other locations on Venus and Earth. Specifically, how “can [it] be inferred” that individual dykes are laterally emplaced?

REE: There is extensive literature on lateral emplacement of regional (mafic) dyke swarms on Earth, Venus and Mars from a variety of scientific constraints including on the basis of theoretical modelling, magnetic and petrographic fabric analysis of samples, geochemistry and paleomagnetic studies along individual dykes. I could easily add many more references in support of this consensus view.

How was this done for the radiating swarm around Artemis? Can those same methods be applied here?

REE: Actually, the interpretation of lateral emplacement of the giant radiating swarm associated with Artemis (Hansen and Olive 2010, *Geology*), does use the same ‘method’ as we have done--- and refers to a number of terrestrial and planetary papers—including several from the Corresponding Author for this GDAR manuscript, Ernst.

Incidentally, this is also the place in the manuscript that highlights why calling the GDAR “longest” might seem like puffery—dykes around Artemis could be much longer.

REE: We are very precise in our statement that this individual graben (interpreted as overlying a dyke) is TRACED for such a distance, giving it the current record for maximum TRACED length of a individual dyke. We certainly anticipate that our record may be broken by similar studies on other long swarms. For instance, the Artemis dykes extend from the rim (at 1000 km radius) out to 6000 km, meaning that individual Artemis dykes have the POTENTIAL to be directly INDIVIDUALLY TRACED up to 5000 km in length. But until this is successfully done, then our GDAR does hold the length record.

- Lines 80–81: The claim that “Atla Regio is recognized as a major plume centre” needs a reference (not a self-citation, please) and elaboration. On what basis do folks infer the presence of a mantle plume? Are there competing hypotheses?

REE: Reference added to key newly-accepted paper in *Commun Earth and Environment* (El Bilali et al. 2023). This reference summarizes the existing literature and provides new conclusive evidence from the radiating graben systems in support of a plume origin for Atla Regio.

- Line 98: “72 km”?

REE: thanks--- the missing “km” has been added

- Line 106–107: “The recognition of the GDAR” is stated as a fact here (and elsewhere) and used to support many further claims. However, the “GDAR” has several breaks in continuity, and the authors have not mustered sufficient proof that these breaks are not real gaps between different dykes. In any case, all these statements need caveats. If the GDAR is one continuous dyke, then we can make certain claims.

REE: It is so well established in the terrestrial literature (since initial key papers of Halls et al.. 1982, and Fahrig, 1985) that major radiating swarms are laterally emplaced. This requires a lateral continuity of dykes (which has been supported in the terrestrial literature, from paleomagnetic and geochemical arguments). Even the observation of en echelon offsets is not an issue with interpreted continuity of dyke at depth (as referenced in our manuscript). See the text below for a more complete discussion of the reasons that dykes of radiating swarms are laterally emplaced (except near the plume centre region).

We could include such text (with references) as a Supplementary File if recommended by the Editor.

Actually, there is much evidence AGAINST vertical emplacement and IN FAVOUR of lateral emplacement in giant radiating swarms.

- 1) There is no geophysical evidence for a widespread magma chamber underneath the entire extent of radiating swarms (the extent of a magmatic underplate can be recognized from geophysics, is only the central part of radiating swarms.--- for some swarms that would require a magma reservoir over millions of square km at the base of the crust.
- 2) Another issue is that many swarms are emplaced in crustal blocks with thick lithospheric roots—sufficiently thick that mafic magma could not be generated from beneath--- the simplest solution is lateral emplacement of the magma from regions of thinner lithosphere at the edge of the craton via lateral dyke emplacement
- 3) Geochemical study indicates separate dykes of a swarm can have a distinct incompatible trace element chemistry along their length, but the unique incompatible trace element chemistry characteristic of each dyke is different from that of nearby dykes of the same swarm. A good example is the 700 km long Great Abitibi dyke of the Canadian shield and its companion long dykes (hundreds of km in length which are separate from the Great Abitibi dyke by only 10s of km. This implies that each individual dyke represents a single distinct magmatic pulse that could extend 100s of km long but be different than the geochemical pulse that is characteristic of a neighboring dyke 100s of km in length. How is this possible to producing such consistent chemistry for linear features (dykes 100s of km along), but be different for adjacent linear features (dykes) in the swarm. This is not consistent with derivation of a giant radiating swarm from a coextensive underlying magma chamber
- 4) Magmatic fabric studies in radiating swarms typically indicate lateral emplacement except near the plume centre
- 5) The surface expression of radiating dyke swarms on Venus and Mars is as graben--- meaning that the magma is not vertically emplaced except locally.

REE: So if a major (distinctive) individual dyke ends and then reappears further along trend, that can only mean (based on the evidence of lateral emplacement of regional-scale radiating dyke swarms) that the intervening portion of the dyke has been obscured—by younger units.

REE: Our argument is not circular----given that we can see the interpreted same major graben (dyke) reappearing further away. Key to this argument is distinct width and appearance of the GDAR graben (see images in Figure 1) which is so unlike any of the other nearby graben.

- Line 126: “Confirming” seems way too strong here. Maybe “consistent with...”? Determining the plume head size would probably require geophysical data like seismology, high-resolution gravity, etc.

REE: We have changed to “consistent with” Again our newly published paper on the Atla Regio Superplume (El Bilali et al., 2023, Commun: Earth and Environment) fully answers this question of plume size of Ozza Mons

- Line 137: Again, “the remarkable continuity of the GDAR provides a conclusive link” is way too strong. It’s not entirely continuous.

REE: We have commented on this above.

- Lines 142–152: Again, the topic sentence of this paragraph is way too strong. Also, “GADR”? The dyke could be used as a precise time marker if it were continuous. However, that seems to be an *assumption* because there is no specific proof that the breaks in visible continuity do not correspond to the real end of a dyke.

REE: The key criterion for assessing dyke continuity is this particular graben is so distinct (by its width and appearance) that it can be reliably traced across gaps (even the 200 km gap). We would be happy have a Zoom call to ‘walk’ along the dyke using the Magellan SAR image and show that a single dyke is a reasonable interpretation.

REE: By the way, my PhD was on the Great Abitibi Dyke (GAD) that we traced for 700 km using both geology maps and aeromagnetic images. However, in a few areas the aeromagnetic maps were busy, leading to small breaks in the continuity of GAD. But since this dyke (GAD) was so wide (90-250 m in thickness) then it was confidently traced across these gaps. Furthermore, geochemistry and paleomagnetic data confirm that the Great Abitibi Dyke (despite some aeromagnetic gaps) was indeed a single dyke: chemistry showed a distinct trace element geochemistry at sites along the dyke (which was different from adjacent dykes) and similarly the paleomagnetic data showed a distinct palaeomagnetic direction along the dyke that differed from that along other adjacent dykes of the swarm (representing emplacement and cooling of the Great Abitibi dyke as a single event in a short time (<1000 years))

Line 145 says that “the only significant break in the continuity of the GDAR is for 200 km” but “significant” is in the eye of the beholder (and I believe disallowed in Nature journals unless one has performed a statistical test with a quantitative threshold). Continuity is the sine qua non of this manuscript, so I believe that proving the stated conclusions would require evaluating every single break in continuity observed along the trace.

REE: We have replaced the word “significant” per the guidance in Nature journals, in which this term should only be used in a statistical sense. In the text we have added more information on each of the breaks (and giving the coordinates of the start and end of each break)

- Lines 160–161: “the surface expression... is visible unless covered by younger flows” implies that the depth of the dyke does not change along its length, correct? If the dyke were vertically emplaced or

laterally emplaced but at different depths along track, then the surface expression could change even if the surface were everywhere the same age?

REE: current understanding of large radiating dyke swarms, indicates that they are emplaced vertically near the plume centre (within a few hundred km) but then emplaced laterally out to the full extent of the swarm, e.g. out to 2300 km in the Mackenzie dyke swarm of Canada (e.g. Ernst and Baragar 1992, Nature, and other more recent references)--- see my more detailed comment above.

REE: however, dominant vertical emplacement may occur in linear swarms, particularly those associated with a rift zone--- but lateral emplacement dominates in giant radiating swarms

REE: There is very little known about the surface expression of dykes on Earth because the surface graben are only rarely observed (due to widespread erosion on Earth). So we don't really know the magma level (depth below surface) on Earth nor Venus. We can speculate (as done by Baragar et al. 1996, J. Petrol.) that dykes will be travelling laterally away from the plume centre and perhaps at a relatively constant level below the surface. In any case, further work that analyzes the width and other characteristics of graben above the GADR on Venus can potentially help address the above question.

We have added some text with some estimate of the characteristic width of the GDAR graben and estimates of the depth to the top of the dyke

- Lines 162–165: Apologies that I sound like a broken record, but I do not believe that it is possible to draw such a strong conclusion from the evidence presented in this manuscript. Specifically, the authors state “nowhere along its 3700 km length did any lava flow spill out of the GDAR. Our conclusion is that the entire volume of magma injected laterally to form the GDAR is still preserved within the dyke.”

REE: But this is exactly what we expect on Earth—part of the recognition of long distance lateral emplacement of dykes on Earth is assuming that the magma is not spilling out. If magma does spill out then this would likely cause the dyke to end or reduce the distance that the dyke could continue to propagate laterally.

Let me draw a qualitative analogy. Baltis Vallis is the longest lava (almost certainly lava) channel on Venus. However, scientists debate whether lava flowed from north to south or from south to north. That's because it's not obvious where lava spilled out of Baltis Vallis. However, I would not thus conclude that all the lava that formed Baltis Vallis is *still inside* Baltis Vallis. Our available data are limited (e.g., the radar images have low-resolution; we do not have compositional data; we haven't yet done sub-surface radar sounding that could identify individual flows). I could make similar arguments about over a hundred other lava channels on Venus. So, can the authors prove that *their failure to observe* lava spilling out of the dyke means that *no lava at all* spilled out of the dyke? Surely there is some amount of lava (maybe a huge amount) that could have spilled out but escaped detection? Maybe EnVision and/or VERITAS will find some? In an extreme case, perhaps nearly the *entire* volume of lava spilled out of the end of the dyke, but then the end of the dyke was covered by a younger flow? Overall, the stated conclusion seems like an extreme upper bound on reality—and the manuscript should acknowledge other possibilities.

REE: Yes, it is our robust conclusion that along its traced length that the GDAR does not spill any lava. We have a lot of experience identifying flows coming out of graben (dykes) on Venus. Our

International Venus Research Group has about 40 students from 5 countries doing detailed mapping of lava flows and grabens all over Venus, having produced 28 abstracts for the Lunar and Planetary Science Conference (LPSC) in 2023 and 34 abstracts for LPSC 2022. So, we can easily recognize flows fed from grabens. Thus, it is remarkable to us that the GDAR does not anywhere along its traced length exhibit flows sourced from the grabens. Also, we do not see any flow emanating from the end of the dyke. We maintain that these are robust observations. This also could be demonstrated via a Zoom tour of Magellan SAR images along the dyke.

REE: Reviewer 3 does make the point that the dyke could actually extend further and feed flows; but that these are obscured by younger cover. We did look for a considerable distance beyond the traced end of the GDAR for any resumption of this dyke—so we think that we have indeed reached the end of the GDAR.

REE: It certainly is also possible that in the 200 km gap where we interpret the dyke to be overlain by a younger flow unit, that there is lava-spilling from this hidden portion of the dyke.. But since along nearly 3500 km of traced dyke we see no spilling, then it is more probable that there is no spilling in the 200 km long hidden portion of the GDAR. In any case we have added text to indicate and acknowledge that we can prove the absence of feeding of flows in the gaps

REE: this analysis of long lava channels (canali) is sensible but is not a relevant analogy to dykes.

- Lines 165–184: Here is the quantitative part of the manuscript, which underlies the central conclusion that the GDAR was fed from a buffered magma reservoir. However, once again, this conclusion depends on an assumption that was not well-justified. The calculation of the magma volume includes an uncertainty analysis related to measurements of the width of the dyke. However, the vertical extent of the dyke is even more uncertain—but no such uncertainty is included in the calculation. Specifically, the manuscript uses 20 km as the vertical extent. If the vertical extent is <10 km, then only one cycle of magma from the reservoir is (maybe) required to fill the dyke—meaning that the conclusion of a “buffered magma reservoir” goes away. How do we know that the vertical extent is not <10 km? Cited references 33 and 9 seem to focus on Mars and Earth. Reference 33 admits vertical heights <10 km—and conditions on Venus could be different than on Earth. In particular, the relevant calculation would require assumptions about the density of the magma and the host rocks. The manuscript does not provide estimates for those densities, which must be considered uncertain in the absence of much relevant geophysical data for Venus. Similarly, one number is used for the volume of the source magma reservoir, but that number is uncertain. As with virtually all the conclusions in this manuscript, the scenario described by the authors seems *possible* but is by no means the only possibility that is consistent with the data.

REE: It is correct that there are significant uncertainties in the width and vertical extent of the Dyke and so the volume estimate has large uncertainties. Indeed, we have used the range in plausible values to bracket the volume and conclude a range in refilling from 1 time to 37 times.

- Lines 197–201: To support transparent and reproducible science, I believe that the authors should make available, in an accessible digital format, all the data used to generate the figures in the manuscript (e.g., all traces and topographic profiles).

REE: Our interpretation is entirely reproducible. The Magellan SAR image data are available via the USGS Astropedia site and also on the JMARS site--- and the map shows the location of the line work which can allow anyone to confirm our tracing of the GDAR. We have also added coordinates all the labelled places along the GDAR in Figure 1 (nos. 1, 2, 3 and letters, B, C, D, and E) as well as for the end of the dyke. So any reader with access to JMARS can easily confirm our results

- References. By my count, almost half of the citations are self-citations, which seems a bit excessive. I recommend reconsidering these self-citations. As with the “1 Ga” “[17]” example discussed above, there might be many cases where it would be more appropriate to cite a paper that directly performed the relevant analysis, rather than a study, which included the senior author, that simply mentioned the issue.

REE: we have added more references to other authors. However, it should be noted that many key papers on mafic dyke swarms on Earth and Venus have been done by our co-authors

Reviewer #2 (Remarks to the Author):

Review of Nature Communications Manuscript: The Great Dyke of Atla Region, Venus: The Longest Traced Individual Dyke in the Solar System

General comments:

This manuscript presents a strong case for identifying the longest single dyke in the Solar System on the planet Venus. This identification could have extremely important implications for our interpretation of stress regimes on Venus, as well as our ability to determine relative ages across large regions. This manuscript is well written and lays out its case carefully. However, it would benefit from some slight restructuring and some additional clarification to better support its argument.

REE: we thank this reviewer for providing an annotated PDF with editorial and other suggestions that we have implemented

Specific Comment:

The authors frequently say that they are observing (or mapping, or measuring) dykes when what they really mean is that they are observing graben that they are interpreting to overlie dykes. This is a pretty important distinction to make. The papers the authors are citing to support their interpretation of dykes beneath the graben all assume that it is the emplacement of the dykes that forms the overlying graben. There is still quite a considerable amount of debate about this issue.

REE: additional support of our dyke swarm interpretation of the graben we are mapping in the Atla Regio superplume area comes from the newly published (El Bilali et al. 2023, Communications Earth and Environment,

For all of the literature that supports the contention that graben form above dykes, there is an equal number of papers showing that dykes don't necessarily form overlying graben. Not every graben on Earth has a dyke beneath it, so it is odd that planetary researchers seem to assume that there is a dyke beneath every graben on Venus (or Mars). And there are multiple locations on Earth where exposed cross-sections of dykes show no fracturing (or deformation of any kind) above them. Physical models have also shown that dyking does not necessarily form graben.

REE: the discussion of the argument for interpreting the families of cross cutting graben sets on Venus as overlying dykes is further discussed in the newly published paper (El Bilali et al. 2023, Communications Earth and Environment,). Although generally difficult to observe because of erosion, there are terrestrial examples of graben above dykes on Earth; we could add additional references on this point if necessary---

To be fair, the authors do state that they are making this interpretation in several places in the manuscript. But it doesn't make it any less jarring when they then talk about measuring the length of a dyke when what they really mean is they measured the length of the graben that they are interpreting to overlie a dyke. And to be clear, I am not trying to suggest that there are no dykes beneath the graben radial to Ozza Mons. I just think that the authors could be more precise and accurate in their writing. Such clarity would only strengthen their arguments and support their case for the presence of a 3700 km long dyke on Venus.

REE: We have modified our text to refer to graben and follow this with text such as the following "interpreted to overlie dykes"

Minor Comments:

I have made some suggestions in an attached annotated PDF that I believe would strengthen the paper.

REE: we have accepted the suggested changes in the annotated PDF

Reviewer #3 (Remarks to the Author):

In this short paper, El Bilali & Ernst analyze Magellan radar images of Venus' Atla Regio to map a giant dyke that is described as the longest dyke traced so far on Venus and the entire solar system. The authors demonstrate a length of at least 3700 km and potentially almost 4500 km, which substantially exceeds the longest dyke traces on Earth. The authors trace the dyke based on its graben signature and use elevation data to identify major troughs. I found this letter relevant and interesting to read and it is relatively easy to follow the manuscript even without being an expert on dyke swarms and/or geological mapping. In this sense, I feel this work is of interest for publication in a journal such as Nature Communications.

I have not spotted any major flaw in the way the results are obtained and interpreted. However, I was a little disappointed about the discussion of broader implications and significance of the detection of such a long dyke. Some issues that I would have liked to see discussed a bit more:

1.) Could such a long dyke imply that Venusian dykes may be longer than on Earth's or does it just demonstrate that terrestrial dykes are much more overprinted and we can trace their original length? Are there implications of such a long dyke on the properties and state of the interior (which are thought to be different and probably have been for substantial time for Earth and Venus)? For instance, could the place constraints on local crustal thickness and strength?

2.) What is a feasible tectonic regime that may support such long dykes? Is this more in line with classical end-members such as the stagnant or mobile lid tectonics, or could that be better explained by some more recent suggestions for Venus' tectonic regime, such as the plutonic-squishy lid?

3.) I understand that the 1200-km-distance at which the directional swing of the dyke occurs is here used as a proxy for the source plume head below Ozza Mons, but I would have liked a short discussion whether this indicates an unusually large plume or not, perhaps by comparison to better-studied terrestrial settings. In other words, is a very long dyke indicative of an unusually large or hot plume head? Or is the length of dyke swarm thought to be controlled mostly by other factors?

4.) The last paragraph (line 190++) points towards searching for other long continuous dykes on Venus. But what are regions on Venus which may display such features more likely than others? Also, in what sense may be upcoming Venus missions – Veritas, DaVinci, and EnVision – help to identify more such dykes and/or provide more details about the structure of the GDAR and its implications?

I think, some elaboration of points 1.) - 4.) would improve the manuscript and its impact on the Venus' science community the most, but I do have some additional minor comments, following below. Overall, I think this paper can be made suitable for publication with minor-to-moderate revision.

REE: We appreciate the suggestion of this review, and have added more text on the implications of our discovery, but in the limitations of space we could not go into detail on all the points suggested

Additional comments:

Line 11++: The abstract lacks a statement of the broader significance for our understanding of either dykes or of Venus' tectonics. It would be good to add one if the word limit allows.

REE: We have added some text on the significance of our discovery

Line 17: Remove the comma after "1200 km".

REE: done

Line 29: "It is inferred ..." Inferred based on what? Also, the word "inferred" seemed to appear quite often throughout the manuscript. Please check and think about alternatives.

REE: agreed "Inferred" is used to much and alternative wording are used

Line 40/41: Where has been the longest dyke in the solar system before your detection of GDAR? Readers may wonder about this (well, I did ...).

REE: The longest traced individual dykes were those mentioned in the paragraph. Probably the longest TRACED individual dyke was 1000 km (Red Sea), but the longest INFERRED length assuming lateral emplacement on Earth is probably the Central Atlantic radiating swarm with dykes going out to 2800 km away from the plume centre (hence the individual (laterally emplaced) dykes that reach the end of the swarm might be up to 2800 km long. The longest on Mars is likely about 4000 km (based on the maximum length of the Tharsis radiating graben systems (e.g. Mege and Ernst 2001). On Venus the largest known swarm is the Artemis radiating swarm which has dykes extending out to nearly 6000 km, but since they appear to start at the edge of the Artemis corona (radius 1000 km) then may be starting from magma reservoirs at the edge of the corona-- in which case their maximum length could be about 5000 km.

Line 57: "...at least back to about 1 Ga ..." Why 1 Ga? Is that supposed to be linked to Venus' mean surface age? This is unknown and, yes, 1 Ga is in the range of estimates. But at the very upper end, and the most recent estimates tend to suggest younger surface ages perhaps as young as 100-200 Myr.

Please clarify, why you write “at least 1 Ga” if actually have very little clue of how Venus’ operated at this time (it may or may not had more active tectonics etc.)

REE: good point and also made by reviewer 1. We are adding these estimates of these ages, citing additional references.

Line 59: “... inferred to overlie blind dykes ...” I do not understand how such blind dykes can be recognized from radar images such as those used here? Would that not need some rather high-resolution gravity data at least? Or is this inference actually just used in analogy to terrestrial dykes for we may know about these blind dykes?

REE: yes, this is reference to the model (applied to both Mars, Venus and Earth) in which graben are inferred to overlie blind dykes. The graben sets (radiating, circumferential and linear) on Venus interpreted to overlie dykes for all the reasons given in the cited references and the most relevant manuscript is our just-published paper on the Atla Regio radiating dyke swarms (El Bilali et al. 2023, Commun: Earth and Environment)

Line 64: I am puzzle by the word “once-compressed”. Could it be also twice-compressed? Does it effectively mean that the compressed images have coarser resolution (225m/px) than the uncompressed (75m/px)?

REE: yes, this is the term that was used for these lower resolution datasets. We have provided a reference for these once-compressed data

Line 69: “In this study [20] ...” should be “In that study [20] ...”!?

REE: Thanks – I have made that correction.

Line 185++: This short paragraph falls a bit short compared to the others. Would you expect similarly long dykes extending outwards from Ozza Mons in various directions (at least until the 1200 km-limit up to which the regional stresses seem to be less significant)? Or could there be some preferential directions? In Figure 1, it seems that there is at least one quadrant (SW) of the blue circle without any radiating graben systems (orange line)? Why?

REE: we expect that the swarms are radiating (due to domal uplift above the plume) out to the edge of the plume beyond which they have trend controlled by the regional stress field.

REE: The absence of radiating dykes in the SW quadrant is due to the presence of younger flows of Maat Mons and Unnamed Mons

Line 203++: DOI’s seem to be missing consistently in the reference. These are always useful to have. It seems references 4 and particularly 30 are earlier (reduced) versions of the contents presented here, so consider deleting these ref’s, especially as they are conference abstracts without peer-review.

REE: Reference 4 has now been replaced by the newly published El Biliali et al. (2023)

REE: Yes, reference 30 can be dropped.

REE: regarding adding the DOIs. I agree they are useful, and we will add them if the journal requests.

REE: We have added and removed some references and renumbered accordingly

Figure 1: I am not sure what the color coding in the large-scale inset (upper right corner of the figure) is meant to represent!?

REE: thanks for catching that. We have added text to the caption to explain the colour coding in the inset

Figure 2: It was not clear to me immediately what the 500 m label in the left panel is supposed to mean. While I got it on a second look, consider adding a note in the caption to make it clear right away. Also, it seems that you show here a ~100-200 km long section from the beginning of the GDAR, that is close to Ozza Mons, I suppose. It would be interesting to see, for comparison, how this compares to a section further away, perhaps at around the 1200-km boundary or even further out. This could potentially indicate structural changes along the dyke

REE: In response to this reviewer comment we have experimented with adding an image closer to the centre, but in this area the geology seems very complicated with multiple graben and areas that are covered by younger flows. There is no obvious way to trace the GDAR closer to the plume centre. We are planning to undertake a more detailed mapping of the central region of Ozza Mons (including the flows) and perhaps that will yield some insights on this point, for a future paper.

REVIEWER COMMENTS

Reviewer #1 (Remarks to the Author):

Re-review of “Far-Travelled (4000 km) Magma Propagation [sic?] Just Below the Surface of Venus” for Nature Communications. September 26, 2023.

I reviewed the new manuscript and the authors’ response to the original reviews. I also read the new manuscript by the authors: “Dyke swarms record the plume stage evolution of the Atla Regio superplume on Venus” by El Bilali et al. (2023) in Communications Earth & Environment.

I leave it to the editors to judge the overall significance of this manuscript, especially given the very similar results presented in El Bilali et al. (2023). Regardless, I have many technical concerns with the manuscript that I believe should be addressed regardless of where it is ultimately published. Some of these concerns are carried over from my first review, but several of my comments relate to new text and numbers that were added for this submission.

Comments (in order of appearance):

- Should “Propagation” be “Propagation” in the title of this manuscript?
- Lines 16, 46–48: I think these sentences need a “so far” or a comparable caveat to avoid giving unwary readers the impression that all long graben on Venus have been traced.
- Figure 1: The subpanel in the top-right corner is basically identical to Figure 1d in El Bilali et al. (2023). In my opinion, this duplication should be acknowledged.
- Lines 70–71: The statement “wind erosion is minor” is rather vague—“minor” by what metric? Overall, the extent of sedimentary activity on the surface of Venus is debated but could be substantial. Even if wind speeds are low, the thick atmosphere can transport small particles. Mass wasting could be common on the steep slopes of graben.
- Line 118: The manuscript asserts “the continuity of the GDR across the gap is unambiguous (based on matching of trend and width of the graben).” The response to reviewers likewise states that “this particular graben is so distinct (by its width and appearance) that it can be reliably traced across gaps.” I wish these statements were specific and quantitative. How does the width of this graben compare to the distribution of widths observed for nearby graben? As noted in the response, we simply do not have the types of datasets (geochemistry, paleomagnetism, etc.) that have been used to prove the continuity of Great Dykes on Earth.
- Lines 121–123: This scaling ratio has substantial uncertainty. Cited reference 9 adopts the value of 3.5 as a rough average from a study of only two dyke-induced graben in Iceland by Rubin (1992). In that study, the specific values were 1.9 and 4. So, the depth to the underlying dyke could be anywhere from ~250 m to 1,600 km based on the 2 ± 1 km width measured in this study. This uncertainty should propagate through to the next calculation in this paragraph—and to the calculations later in the manuscript.
- Lines 123–125: I am not convinced that this depth estimate is robust. What is the horizontal resolution of the Magellan altimetry along the graben? Is the stereo-derived topography available here? If the

horizontal resolution is much worse than the width of the graben (it might be an order of magnitude worse...), then the vertical resolution of the Magellan altimetry data is not a strict limit. A deeper depth would be “smoothed out” (i.e., look less deep) in poorly resolved altimetry data.

- Lines 188–189: The last sentence of the preceding paragraph seems to contradict the first sentence of this paragraph. That is, I’m not sure that “flow” describes fine-grained material from an impact? So, it seems possible that, besides (lava) flows, sediment (which can form on Venus from impact or other processes) could pile up in certain locations and obscure graben.

- Lines 190–192: Again, I believe the word “any” is inappropriate in this sentence. The radar images of Venus are simply insufficient to exclude “any” volume of lava flowing out of the GDAR. In response to my previous comment, the authors pointed out that they have submitted >50 abstracts to the Lunar and Planetary Science Conference. However, I maintain that lava could have spilled out at spatial scales that are not resolvable in the available imagery (e.g., <10,000 m²). Alternatively, the spilled lava could have radar properties that make it difficult to distinguish from the surrounding plains. I believe that the authors “did not observe lava flows spilling out of the GDAR.” Therefore, there is probably a (non-zero!) upper limit to the amount of lava that could have spilled out. They should say that, recognizing that absence of evidence is not always evidence of absence.

- Line 201 states that “Venusian crust is typically 20-25 km thick, thick.” (Maybe one “thick” is enough.) However, Maia & Wiczorek (2022, JGR) found that, under some assumptions, “the global average crustal thickness of Venus is about 20 km.” Previously, James et al. (2013, JGR) found that “the mean thickness of the crust is constrained to a range of 8–25 km.” So, I wouldn’t be surprised if Venusian crust were thinner than considered here, at least over half or more of the surface.

- Lines 226–228: The phrase “will be valuable for contributing to target selection by the fleet of new missions to Venus planned over the next decade (VERITAS, DAVINCI, EnVision, Venera-D, Shukrayaan-1, VOICE)” also appears verbatim in El Bilali et al. (2023). This verges on self-plagiarism. I recommend that the authors double-check the entire manuscript for similar problems.

Reviewer #4 (Remarks to the Author):

In my opinion, this manuscript is interesting, well structured, and the science is quite sound, although I have a few minor concerns that I believe the authors should address before this paper can be published in a high-end journal such as Nature Communications. These minor concerns fall in line with the concerns raised by reviewer #1, whose comments are, in my opinion, often replied to in the “reply to reviewers” but then not thoroughly taken into the revised manuscript, and also (importantly) about the data availability. The detailed review (a couple of points) can be found in the attached PDF.

[Editorial Note: The contents of the PDF have been included below]

This is a very interesting study and detailed mapping of the Venusian surface, with an exciting new feature thoroughly analyzed, and interpreted to be an extremely far-travelled magmatic dyke. As is an external review in the second round of this manuscript’s review, below, I assessed both the manuscript as the comments to concerns raised by previous reviewers.

In my opinion, this manuscript is interesting, well structured, and the science is quite sound, although I have a few minor concerns that I believe the authors should address before this paper can be published in a high-end journal such as Nature Communications. These minor concerns fall in line with the concerns raised by reviewer #1, whose comments are, in my opinion, often replied to in the “reply to reviewers” but then not thoroughly taken into the revised manuscript. I quite am certain that many readers would have similar concerns, even when they appear “stupid” to the authors. The editor and authors should consider the factors summarized here:

- The interpretation of the surface structure as a laterally continuous dyke should be better described, it is not a given. What may seem “obvious” to the author does not come across as such in the manuscript. I would urge the authors to address, in the paper or in a supplement, the details behind recognizing GDAR as a laterally continuous dyke, and to also discuss some alternative interpretations (e.g., as one of the reviewers mentioned, there is a wealth of literature showing that dykes don’t necessarily form overlying graben; not every graben on Earth overlies a dyke).
- Throughout the authors excessively make use of self-citation. I agree that one of the authors has long-standing research output in the field of (dyke) mapping and that many papers on mafic dyke swarms on Venus/Earth include this author. Yet this is not the case in the field of Venus science, tectonics, geodynamics, or past mission observations. Any statement about interpretations of Venusian research should be supported by the original works, not only by the author’s latest paper. I urge the editor to take important note of this.
- I see absolutely no reason for the authors not to accompany the datasets of their mapped structures (e.g., a shape file or other file, to be opened in GIS software or JMARS), and the topographic profiles of the structures, with this manuscript. Especially since these files should not be extremely large or sensitive, I see no reason to leave them out. I do not request this for interpretation reproducibility per se, but for the FAIR data principles that allows for reusability and for other scientists to include the mapped structures in their own future mapping endeavors (with obviously the appropriate references to this paper). Nature’s data ethics include “authors are required to make materials, data, code, and associated protocols promptly available to readers without undue qualifications” and making available the “minimum dataset necessary to extend the research in the article”.

You can find some more details with line numbers below:

Lines 71-72: This sentence is quite complex maybe since it misses a “)”? Also, I would suggest a recent review to be added as reference regarding the present-day knowledge on the surface age of Venus (few 100 Myr to ~1 Gyr, or widely distributed if the surface age is in fact not uniform but formed in a so-called “equilibrium” resurfacing model):

<https://doi.org/10.1007/s11214-023-00966-y>

Lines 95-99 (and concern reviewer #1): the claim that “Atla Regio is recognized as a major plume

center” should have an original citation of key Venus papers rather than a self-citation their recent paper. The earliest and most convincing argument for a plume center beneath Atla Regio (and for that matter, the BAT region) comes from gravity studies that reveal a deep dynamic support at these regions. These original sources have to be cited and ideally, in the paper, it should even be discussed why the Atla Regio is inferred to be a “hotspot” region:

<https://doi.org/10.1111/j.1365-246X.1997.tb00593.x>

<https://doi.org/10.1006/icar.1994.1166>

<https://doi.org/10.1029/95JE01834>

Lines 161-162 “and consistent with a plume head radius for Ozza Mons of 1200 km diameter” this one-on-one correlation comes with assumptions that should be briefly mentioned. This is assuming that the plume head completely reaches the edge of the “purely radiating pattern”. Couldn’t radiating pattern extend far beyond the lateral plume head size (e.g. numerical models)? Please discuss or, if convinced, mention why this one-on-one correlation is correct with appropriate references.

Throughout the paper (and concern reviewer #1) The confident use of the “recognized GDAR” can, in my opinion, only be valid in this manuscript when the authors better address their interpretation and assumption. Not all grabens on Earth overly dykes, so why are you so convinced this one does? Please clarify or add a supplement with a paragraph of details, for the reader who may be unaware of this. Here, the authors can also go in depth on exactly why the minor lateral discontinuities are not affecting their interpretation of the dyke being laterally continuous and the result of lateral emplacement rather than vertical. The authors have replied to this concern in detail (albeit, also dismissive) to reviewer #1 but not so much in the paper.

RESPONSE TO REVIEWS

2023 Dec 23

All responses in red font

RE-REVIEW BY ORIGINAL REVIEWER 1

I reviewed the new manuscript and the authors' response to the original reviews. I also read the new manuscript by the authors: "Dyke swarms record the plume stage evolution of the Atla Regio superplume on Venus" by El Bilali et al. (2023) in Communications Earth & Environment.

I leave it to the editors to judge the overall significance of this manuscript, especially given the very similar results presented in El Bilali et al. (2023).

RESPONSE: We appreciate that the editor assessed that this Great Dyke paper represents a distinct contribution with respect to our earlier paper (El Bilali et al. 2023) on the Atla Regio Superplume

Regardless, I have many technical concerns with the manuscript that I believe should be addressed regardless of where it is ultimately published. Some of these concerns are carried over from my first review, but several of my comments relate to new text and numbers that were added for this submission.

RESPONSE: We have endeavored to accommodate all the comments of this review—see response below

Comments (in order of appearance):

- Should "Propogation" be "Propagation" in the title of this manuscript?

- **RESPONSE:** First of all thank you to the reviewer for catching the spelling error. We have added the word "Lateral " FAR-TRAVELLED (4000 KM) LATERAL MAGMA PROPAGATION JUST BELOW THE SURFACE OF VENUS—Our interpretation is clearer now by adding the word lateral, since our point that such a long single dyke would be emplaced laterally from the plume centre region.

-

- Lines 16, 46–48: I think these sentences need a "so far" or a comparable caveat to avoid giving unwary readers the impression that all long graben on Venus have been traced.

- **RESPONSE:** We thank the reviewer for this suggestion. "so far" has been added.

- Figure 1: The subpanel in the top-right corner is basically identical to Figure 1d in El Bilali et al. (2023). In my opinion, this duplication should be acknowledged.

RESPONSE: we have acknowledged that this subpanel is after El Bilali et al. (2023). This is an ideal location diagram for any part of the BAT region and so we think it makes sense to also use it here.

- Lines 70–71: The statement "wind erosion is minor" is rather vague—"minor" by what metric? Overall, the extent of sedimentary activity on the surface of Venus is debated but could be substantial. Even if wind speeds are low, the thick atmosphere can transport small particles. Mass wasting could be common on the steep slopes of graben.

- **RESPONSE:** Yes, there are a few areas of recognized sand dunes and plenty of examples of mass wasting associated with rift zones, and apart from these local areas of erosion, in a general way the level of erosion has to be minor. The morphology of lava flows is so well preserved in most areas that the cumulative effect of any cumulative erosion has to be minor. There is also the observation of

dark parabolic haloes caused by dust from bolide impacts and these are modelled to persist some 40 myr before they are successfully degraded and removed by wind erosion. We could have added additional references for all these points but this would have added about 5 references and we already needed to add references in response to other reviewer comments.

- Line 118: The manuscript asserts “the continuity of the GDR across the gap is unambiguous (based on matching of trend and width of the graben).” The response to reviewers likewise states that “this particular graben is so distinct (by its width and appearance) that it can be reliably traced across gaps.” I wish these statements were specific and quantitative. How does the width of this graben compare to the distribution of widths observed for nearby graben? As noted in the response, we simply do not have the types of datasets (geochemistry, paleomagnetism, etc.) that have been used to prove the continuity of Great Dykes on Earth.

- **RESPONSE:** This is an easy point to respond to: We have now provided the line work (ARCGIS shape file) that will allow anyone to duplicate our work--- to see exactly how distinctive this particular feature is. We have reiterated that for 1700 km of its length this feature is named as Penthesilea Fossa, fossa (as the singular of fossae) indicating a single graben. This individual graben is traced across Taussig Quadrangle V-39 (see quadrangle map of Brian et al. 2005). Our mapping confirms the observation in Brian et al. (2005) and tracing the single graben outside the side area, to the west to be able to connect it with Ozza Mons. We added new text to make this point.

- Lines 121–123: This scaling ratio has substantial uncertainty. Cited reference 9 adopts the value of 3.5 as a rough average from a study of only two dyke-induced graben in Iceland by Rubin (1992). In that study, the specific values were 1.9 and 4. So, the depth to the underlying dyke could be anywhere from ~250 m to 1,600 km based on the 2 ± 1 km width measured in this study. This uncertainty should propagate through to the next calculation in this paragraph—and to the calculations later in the manuscript.

RESPONSE: We have now used error propagation to develop our final number which indicates a width of 250 +/- 195 m.

- Lines 123–125: I am not convinced that this depth estimate is robust. What is the horizontal resolution of the Magellan altimetry along the graben? Is the stereo-derived topography available here? If the horizontal resolution is much worse than the width of the graben (it might be an order of magnitude worse...), then the vertical resolution of the Magellan altimetry data is not a strict limit. A deeper depth would be “smoothed out” (i.e., look less deep) in poorly resolved altimetry data.

- **RESPONSE:** The stereo topography is not going to help since the vertical resolution of the stereo-topography is still only 100-50 km and with the horizontal resolution having been reduced to a few km rather than 20 km, this will still be insufficient to resolve the depth of the graben. We now acknowledge the uncertainties in our calculation, and make that the point that a similar analysis will be useful when the new Venus missions provide higher resolution altimetry and imagery.

- Lines 188–189: The last sentence of the preceding paragraph seems to contradict the first sentence of this paragraph. That is, I’m not sure that “flow” describes fine-grained material from an impact? So, it seems possible that, besides (lava) flows, sediment (which can form on Venus from impact or other processes) could pile up in certain locations and obscure graben.

RESPONSE: It is widely recognized that graben can be obscured by lots of younger processes: younger lava flows, impact ejecta and dust (the parabolic haloes mentioned above). However, our key point is that the Penthesilea and its continuation to the west (to the inferred source at Ozza

Mons--is that the feature is traceable--- i.e. is therefore NOT covered by any younger material. This is our basic and key observation and is consistent with the cross-cutting relationships that clearly indicate that the GDAR is younger than most features and thus cuts across them.

- Lines 190–192: Again, I believe the word “any” is inappropriate in this sentence. The radar images of Venus are simply insufficient to exclude “any” volume of lava flowing out of the GDAR. In response to my previous comment, the authors pointed out that they have submitted >50 abstracts to the Lunar and Planetary Science Conference. However, I maintain that lava could have spilled out at spatial scales that are not resolvable in the available imagery (e.g., <10,000 m²). Alternatively, the spilled lava could have radar properties that make it difficult to distinguish from the surrounding plains. I believe that the authors “did not observe lava flows spilling out of the GDAR.” Therefore, there is probably a (non-zero!) upper limit to the amount of lava that could have spilled out. They should say that, recognizing that absence of evidence is not always evidence of absence.

- **REPOSNE: We have added this point – that given the pixel size of about 100 m, then a minimum estimate that would be expelled in any one spot that would be unresolvable would be 100 x 100 equals 10,000 km². While acknowledging this point (that we cannot rule out very small volumes being expelled) our main point still stands, that there is no major expulsion of magma observed anywhere along the length of the GDAR**

- Line 201 states that “Venusian crust is typically 20-25 km thick, thick.” (Maybe one “thick” is enough.) However, Maia & Wiczorek (2022, JGR) found that, under some assumptions, “the global average crustal thickness of Venus is about 20 km.” Previously, James et al. (2013, JGR) found that “the mean thickness of the crust is constrained to a range of 8–25 km.” So, I wouldn’t be surprised if Venusian crust were thinner than considered here, at least over half or more of the surface.

- **RESPONSE: We add these other estimates of crustal thickness. However, the thickness of the crust is not the only consideration, as the analysis of Wilson and Head (2002) suggested dyke heights extending through the crust into the mantle lithosphere.**

- Lines 226–228: The phrase “will be valuable for contributing to target selection by the fleet of new missions to Venus planned over the next decade (VERITAS, DAVINCI, EnVision, Venera-D, Shukrayaan-1, VOICE)” also appears verbatim in El Bilali et al. (2023). This verges on self-plagiarism. I recommend that the authors double-check the entire manuscript for similar problems.

RESPONSE: We have edited this sentence to avoid any self-plagiarism.

Reviewer #4 (Remarks to the Author):

In my opinion, this manuscript is interesting, well structured, and the science is quite sound, although I have a few minor concerns that I believe the authors should address before this paper can be published in a high-end journal such as Nature Communications. These minor concerns fall in line with the concerns raised by reviewer #1, whose comments are, in my opinion, often replied to in the “reply to reviewers” but then not thoroughly taken into the revised manuscript, and also (importantly) about the data availability. The detailed review (a couple of points) can be found in the attached PDF.

RESPONSE: We have now fully addressed each point from the re-review of Review 1 above (see above)

and the responded to the specific comments of new Reviewer 4 below

Title: Far-Travelled (4000 km) magma propagation just below the surface of Venus

This is a very interesting study and detailed mapping of the Venusian surface, with an exciting new feature thoroughly analyzed, and interpreted to be an extremely far-travelled magmatic dyke. As is an external review in the second round of this manuscript's review, below, I assessed both the manuscript as the comments to concerns raised by previous reviewers.

RESPONSE: Thank you for the positive comments

In my opinion, this manuscript is interesting, well structured, and the science is quite sound, although I have a few **minor** concerns that I believe the authors should address before this paper can be published in a high-end journal such as Nature Communications. These minor concerns fall in line with the concerns raised by reviewer #1, whose comments are, in my opinion, often replied to in the "reply to reviewers" but then not thoroughly taken into the revised manuscript. I quite am certain that many readers would have similar concerns, even when they appear "stupid" to the authors.

RESPONSE: Thank you for the positive comments. We see the importance of better explaining and demonstrating to the broad readership what we have recognized from our detailed work.

The editor and authors should consider the factors summarized here:

- The interpretation of the surface structure *as a laterally continuous dyke* should be better described, it is not a given. What may seem "obvious" to the author does not come across as such in the manuscript. I would urge the authors to address, in the paper or in a supplement, the details behind recognizing GDAR as a laterally continuous dyke, and to also discuss some alternative interpretations (e.g., as one of the reviewers mentioned, there is a wealth of literature showing that dykes don't necessarily form overlying graben; not every graben on Earth overlies a dyke).

RESPONSE: We have now added a section that fully considers alternative geological interpretations of such an amazing long narrow feature.

- Throughout the authors excessively make use of **self-citation**. I agree that one of the authors has long-standing research output in the field of (dyke) mapping and that many papers on mafic dyke swarms on Venus/Earth include this author. Yet this is not the case in the field of Venus science, tectonics, geodynamics, or past mission observations. Any statement about interpretations of Venusian research should be supported by the original works, not only by the author's latest paper. I urge the editor to take important note of this.

RESPONSE: We acknowledge this concern and have now added more referencing to the original work, and only cited our work where our contribution was relevant and significant.

- I see **absolutely no reason** for the authors not to accompany the datasets of their mapped structures (e.g., a shape file or other file, to be opened in GIS software or JMARS), and the topographic profiles of the structures, with this manuscript. Especially since these files should not be extremely large or sensitive, I see no reason to leave them out. I do not request this for interpretation reproducibility per se, but for the FAIR data principles that allows for **reusability** and for other scientists to include the mapped structures in their own future mapping endeavors (with obviously the appropriate references to this paper). Nature's data ethics include "*authors are required to make materials, data, code, and associated protocols promptly available to readers without undue qualifications*" and making available the "*minimum dataset necessary to extend the research in the article*".

RESPONSE: We have now included the ArcGIS shape files for that the linework shown. The generalized linework for the Oza Mons system and the detailed linework for the GDAR. This will allow any reader to

reproduced our mapping. We are not adding the topographic profiles since the profiles were generated in the on-line software JMARS and we exported the profile images—and there were no data tables to export.

You can find some more details with line numbers below:

Lines 71-72: This sentence is quite complex maybe since it misses a “)”?

RESPONSE: Parenthesis added

Also, I would suggest a recent

review to be added as reference regarding the present-day knowledge on the surface age of Venus (few 100 Myr to ~1 Gyr, or widely distributed if the surface age is in fact not uniform but formed in a so-called

“equilibrium” resurfacing model):

<https://doi.org/10.1007/s11214-023-00966-y>

RESPONSE: We have added this reference, as a useful review, but the timing of resurfacing is not important to our paper—the key point is the absence of erosion which indicates why we see the surface expression of dykes.

Lines 95-99 (and concern reviewer #1): the claim that “Atla Regio is recognized as a major plume center” should have an original citation of key Venus papers rather than a self-citation their recent paper.

The earliest and most convincing argument for a plume center beneath Atla Regio (and for that matter, the

BAT region) comes from gravity studies that reveal a deep dynamic support at these regions. These original

sources have to be cited and ideally, in the paper, it should even be discussed why the Atla Regio is inferred

to be a “hotspot” region:

<https://doi.org/10.1111/j.1365-246X.1997.tb00593.x>

<https://doi.org/10.1006/icar.1994.1166>

<https://doi.org/10.1029/95JE01834>

RESPONSE: we have added the Smrekar and Stofan references but also kept El Bilali et al. 2023 since that provides new insights into the specifics of the plume history

Lines 161-162 “and consistent with a plume head radius for Ozza Mons of 1200 km diameter”

this one-on-one correlation comes with assumptions that should be briefly mentioned. This is assuming that the plume head completely reaches the edge of the “purely radiating pattern”. Couldn’t radiating pattern extend far beyond the lateral plume head size (e.g. numerical models)? Please discuss or, if convinced, mention why this one-on-one correlation is correct with appropriate references.

RESPONSE: We can now reference the paper by Chadda et al. 2024 (Icarus) in which we discuss the basis for this interpretation.

Throughout the paper (and concern reviewer #1) The confident use of the “recognized GDAR” can, in my opinion, only be valid in this manuscript when the authors better address their interpretation and assumption. Not all grabens on Earth overly dykes, so why are you so convinced this one does? Please clarify or add a supplement with a paragraph of details, for the reader who may be unaware of this. Here,

the authors can also go in depth on exactly why the minor lateral discontinuities are not affecting their interpretation of the dyke being laterally continuous and the result of lateral emplacement rather than vertical. The authors have replied to this concern in detail (albeit, also dismissive) to reviewer #1 but not so much in the paper.

RESPONSE: We have expanded the text that discusses the reasons that Venusian graben are typically thought to overlie dykes. We have also included a specific new section that discuss the GDAR graben and all possible origins: fault (strike slip or vertical), or zone of pure extension, before concluding that underlying dyke is most logical. Also we have discussed why the dyke composition should be mafic and also why an interpretation of lateral emplacement of the dyke from the plume centre region is reasonable.

REVIEWERS' COMMENTS

Reviewer #4 (Remarks to the Author):

I believe that the authors have responded to my comments quite well:

- (a) They have added more original references (although some are in my opinion missing, see below)
- (b) They made the source data available
- (c) They included a much better discussion for their interpretation of a GDAR

Some minor comments (some with a reoccurring theme of erosion) prevail, that are stated below:

Line 71 “Due to the current hot atmosphere (ca. 450° C)” note that only the lowermost atmosphere/surface temperature reaches this value, and that there is a steep gradient in temperature only at the bottom few 10km of the atmosphere. This is confusing, I’d suggest replacing “hot atmosphere” by “hot surface temperature”

Line 72: this relates to my point in my previous review about lines 71-72 about adding the reference <https://doi.org/10.1007/s11214-023-00966-y>. The authors claim they have added this reference but instead I see a reference to an older paper. I suggest to still include (or replace) this more recent reference since it provides a better overview of our present-day knowledge of Venus’ resurfacing, several decades after Magellan.

Line 78 “The absence of erosion” This is contradicting with the previous paragraph: you argue that fluvial erosion is absent, and that wind erosion is minor (but still present). As discussed in Carter et al. (2023) <https://doi.org/10.1007/s11214-023-01033-2> (which in my opinion also needs to be cited for here and lines 71-72) , erosion is indeed present in various forms on Venus. Please rephrase (e.g. “absence” into “relatively minor role”

Line 118 and 223: Similar comment on “absence/lack of erosion”: erosion may be minor compared to on Earth, but it is not 100% absent (see Carter et al. 2023). Rephrase into either something more nuanced or suggestive.

Minor editorial comments (could also be handled by editorial team):

Line 160: I suggest replacing “so” by “therefore”

Line 180 odd phrasing. I suggest “on this basis is linked to” to be rephrased to “is therefore linked to”

Line 181: Odd long sentence, I suggest to break this sentence and start a new one after “Ozza Mons”.

Lines 249-250: Could the authors make one continuously flowing sentence of this? The two different sentences read a bit odd. Suggestion: something along the lines of “Estimates for the thickness of Venusian crust vary, and average values typically vary between 8-25 km [53,54,55], with larger values for

highlands [e.g. 54]”.

Line 271: Remove “Basically” and define “it”

RESPONSE TO REVIEWERS: IN RED FONT
(2024 January 21)

REVIEWERS' COMMENTS

Reviewer #4 (Remarks to the Author):

I believe that the authors have responded to my comments quite well:

RESPONSE: Thank you

(a) They have added more original references (although some are in my opinion missing, see below)

(b) They made the source data available

(c) They included a much better discussion for their interpretation of a GDAR

Some minor comments (some with a reoccurring theme of erosion) prevail, that are stated below:

Line 71 “Due to the current hot atmosphere (ca. 450° C)” note that only the lowermost atmosphere/surface temperature reaches this value, and that there is a steep gradient in temperature only at the bottom few 10km of the atmosphere. This is confusing, I’d suggest replacing “hot atmosphere” by “hot surface temperature”

RESPONSE: Revised as suggested

Line 72: this relates to my point in my previous review about lines 71-72 about adding the reference <https://doi.org/10.1007/s11214-023-00966-y>. The authors claim they have added this reference but instead I see a reference to an older paper. I suggest to still include (or replace) this more recent reference since it provides a better overview of our present-day knowledge of Venus’ resurfacing, several decades after Magellan.

RESPONSE: the new Carter et al. (2023) paper has been added here, but the Herrick et al. (2023) resurfacing paper was already added later in the sentence

Line 78 “The absence of erosion” This is contradicting with the previous paragraph: you argue that fluvial erosion is absent, and that wind erosion is minor (but still present). As discussed in Carter et al. (2023) <https://doi.org/10.1007/s11214-023-01033-2> (which in my opinion also needs to be cited for here and lines 71-72) , erosion is indeed present in various forms on Venus. Please rephrase (e.g. “absence” into “relatively minor role”

RESPONSE: the excellent Carter et al. (2023) reference has been added. Thank you for bringing this reference to our attention. We have made the suggested edit of replacing “absence” with “relatively minor role”

Line 118 and 223: Similar comment on “absence/lack of erosion”: erosion may be minor compared to on Earth, but it is not 100% absent (see Carter et al. 2023). Rephrase into either something more nuanced or suggestive.

RESPONSE: change to “minor erosion”, and “lack of significant erosion”

Minor editorial comments (could also be handled by editorial team):

Line 160: I suggest replacing “so” by “therefore”

RESPONSE: Revised as suggested

Line 180 odd phrasing. I suggest “on this basis is linked to” to be rephrased to “is therefore linked to”

RESPONSE: Revised as suggested

Line 181: Odd long sentence, I suggest to break this sentence and start a new one after “Ozza Mons”.

RESPONSE: This section is rewritten for improved clarity and flow

Lines 249-250: Could the authors make one continuously flowing sentence of this? The two different sentences read a bit odd. Suggestion: something along the lines of “Estimates for the thickness of Venusian crust vary, and average values typically vary between 8-25 km [53,54,55], with larger values for highlands [e.g. 54]”.

RESPONSE: Revised as suggested

Line 271: Remove “Basically” and define “it”

RESPONSE: Revised as suggested